# Commercial Microwave Links as a tool for operational rainfall monitoring in Northern Italy

Giacomo Roversi[1], Pier Paolo Alberoni[2], Anna Fornasiero[2], and Federico Porcù[1]

[1]Department of Physics and Astronomy, University of Bologna, Bologna, 40100, Italy
[2]Arpae-SIMC, Bologna, 40100, Italy

**Correspondence:** Federico Porcù (federico.porcu@unibo.it)

**Abstract.** There is a growing interest in emerging opportunistic sensors for precipitation, motivated by the need to improve its quantitative estimates at the ground. The scope of this work is to present a preliminary assessment of the accuracy of CMLs retrieved rainfall rates in Northern Italy. The CML product, obtained by the open-source RAINLINK software package, is evaluated on different scales (single link, 5km×5km grid, river basin) against the precipitation products operationally used at Arpae-SIMC, the Regional Weather Service of Emilia-Romagna, in northern Italy. The results of the 15 min single-link validation with close-by rain gauges show high variability, which can be caused by the complex area physiography and precipitation patterns. Known sources of errors (e.g. the attenuation caused by the wetting of the antennas or random fluctuations in the baseline) are particularly hard to mitigate in these conditions without a specific calibration, which has not been implemented. However, hourly cumulated spatially interpolated CML rainfall maps, validated with respect to the established regional gauge-based reference, show similar performance ($R^2$ of 0.46 and CV of 0.78) to adjusted radar-based precipitation gridded products and better than satellite- ones. Performance improves when basin-scale total precipitation amounts are considered ($R^2$ of 0.83 and CV of 0.48). Avoiding regional-specific calibration therefore does not preclude the algorithm from working but has some limitations in POD and accuracy. A widespread underestimation is evident at both grid box (Mean Error of -0.26) and basin-scale (Multiplicative Bias of 0.7), while the number of false alarms is generally low and gets even lower as link coverage increases. Taking into account also delays in the availability of the data (latency of 0.33 hours for CML against 1 hour for the adjusted radar and 24h for the quality-controlled rain gauges), CML appears as a valuable data source in particular from a local operational framework perspective. Finally, results show complementary strengths for CMLs and radars, encouraging joint exploitation.

## 1 Introduction

High spatial and temporal variability make precipitation one of the most difficult geophysical observables to measure and monitor. Its accurate measurement would benefit a wide range of applications in meteorology, hydrology, climatology, agri-

culture, just to name the most direct fields where rainfall plays a key role. The precipitation rate can be measured or estimated directly at the ground or using different remote sensing approaches. Rain gauge networks provide point-like measurements of the amount of rain fallen within the instrument's sampling area, cumulated over time intervals which usually range from one minute to one day, with well known instrumental constraints (Lanza and Stagi, 2012) and representativeness limitations (Porcù et al., 2014). Ground-based weather radars, often deployed in large scale networks (Serafin and Wilson, 2000; Huuskonen et al., 2014; Saltikoff et al., 2019), are widely used by hydro-meteorological services to quantitatively monitor precipitation fields, being an effective trade off between spatial-temporal coverage and accuracy in the measurements. However, radar estimates are affected by several errors, which the last generation polarimetric systems have only partially mitigated (Figueras i Ventura et al., 2012; Gou et al., 2019). Satellite estimates received a renewed boost in the last decade from the full exploitation of the Global Precipitation Measurement mission (GPM, Skofronick-Jackson et al. (2017)) that operationally releases a new suite of precipitation products with a high temporal and spatial resolution (Mugnai et al., 2013; Grecu et al., 2016). Despite the undoubted potentials of satellite products to provide estimates over open oceans and regions not equipped with ground instruments, their accuracy is difficult to assess at high spatial and temporal scales (Tang et al., 2020), and their latency hinders the use in real-time monitoring of rain patterns.

A relatively new and independent approach to the estimate of precipitation at the ground became available in the last decades thanks to the growing number of microwave links (or Commercial Microwave Links, CMLs) employed for the backhauling of the cellular communication networks. Growth which only recently and only in some densely populated areas seems to have come to a halt. Integrated precipitation content along a straight path between two antennas can be estimated by measuring the attenuation of the microwave signal travelling down the same path (Turner and Turner, 1970; Harden et al., 1978). Accurate experiments with dedicated hardware and numerical simulation were used to assess the capability of microwave links to measure average rainfall rates (Rahimi et al., 2003), drop size distribution (Rincon and Lang, 2002; van Leth et al., 2019) and water content (Jameson, 1993). The possibility to have a spatially continuous rainfall field depends on the density and distribution of the links, making the CML approach of particular interest for urban areas (Upton et al., 2005; Overeem et al., 2011; Fenicia et al., 2012; Fencl et al., 2013; Rios Gaona et al., 2017; de Vos et al., 2018) with also direct hydrological use in combination with conventional instruments (Grum et al., 2005; Fencl et al., 2013). A further application of the CML approach could be in regions where other instruments are lacking or absent at all (Mulangu and Afullo, 2009; Abdulrahman et al., 2011; Doumounia et al., 2014). However, as it happens for conventional precipitation instruments, the quality of the retrieval is sensitive to several factors, which are often difficult to control (Leijnse et al., 2008), and to the precipitation microphysical structure (Berne and Uijlenhoet, 2007; Leijnse et al., 2010). Given these limitations intrinsic to the measurement geometry and to the nature of precipitation, possible synergistic approaches are considered, to minimize the uncertainties of the different instruments, suggesting the blending of CML measurements with conventional precipitation estimates, such as rain gauges (Fencl et al., 2017; Haese et al., 2017), radar (Cummings et al., 2009; de Vos et al., 2019), or both (Grum et al., 2005; Bianchi et al., 2013).

Even if the general relationship between signal attenuation and rain rate is already well established, the successful use of CML data to quantitatively monitor precipitation still depends on the quality and technical characteristics of the transmitted power data and on the fine-tuning of the algorithms. The somewhat standardized policies of acquisition and storage of the

different companies in different countries make the use of CMLs feasible all around the world, but there is no standard way yet to access them as scientific data. As they consist mostly of confidential maintenance data, major obstacles to face are the widespread unwillingness of releasing them cost-free and the inadequate data-quality standards (Chwala and Kunstmann, 2019).

The first objective of the present work is to make a validation of the precipitation amounts and distributions estimated only from CML attenuation data, using a well-established, freely-available algorithm (i.e. RAINLINK, Overeem et al. (2016a)), over two areas of interest in the Po Valley (provinces of Bologna and Parma), where CML data have been obtained from Vodafone (by direct purchase). Both areas contain river basins of considerable local interest, which will be explicitly addressed. Moreover, we consider for intercomparison only precipitation products routinely available at Meteorological Service of the Regional Agency for Environmental Protection and Energy (Arpae-SIMC): this, on the one hand, prevents us from performing a solid calibration of the algorithm (see Section 3.2), but, on the other hand, allows us to understand how CML product behaves with respect to other operational products. The further aim of the validation study is thus to test the potential of the technology even at its most basic implementation, indicating where to direct the tuning efforts, to set the background for possible inclusion of CML data in the operational routine procedures for precipitation monitoring.

In Section 2 we will describe the area of interest and the different rainfall datasets (CML, radar and rain gauges), including data quality and coverage. In Section 3 we will describe the RAINLINK algorithm and discuss its application to the Emilia-Romagna area. The comparison – at single link and gridded map scales – between the rainfall estimates from the different data sources is presented in Section 4 and discussed in Section 5, while conclusions are provided in Section 6.

## 2  Data

We have considered 57 days from 5 May to 30 June 2016. The two target areas for which we have available CML data are the provinces of Bologna (BO, 3702 km$^2$) and Parma (PR, 3447 km$^2$), both in the Po Valley in Emilia-Romagna, northern Italy (coloured areas in Figure 1). The physiography of the two regions is similar: the highest peaks (about 1,500 m a.s.l.) are located on the southern border, in the Apennine chain, while the central and northern part of the two areas are flat land. The two river basins (thick lines in Figure 1) are both located in the hilly region and have their closing sections located near the cities, in densely populated and assets-rich areas.

Precipitation climatology in the Po Valley during the late spring season is characterized by both stratified structures and small scale convection, with the maxima of the rainfall amounts located on the Apennines ridge (see Supplement). We divided the whole area into square boxes of 5km×5km (see also Section 2.2.2) and this grid will be used to carry out rainfall interpolation and products intercomparison.

The validation has been carried out comparing, at different spatial and temporal scales, the rain amount obtained by CMLs, through the RAINLINK algorithm (Overeem et al., 2016a) with other rainfall estimates operationally available over the target

domain. In particular, CML product has been compared with radar surface rain rates, both raw and gauge-corrected, rain gauges measurements and the operational precipitation analysis (ERG5) made available by Arpae-SIMC.

## 2.1 CMLs

Microwave attenuation data and metadata were purchased as a single dataset of two months, from Vodafone Italia S.p.A. within the Life EU project called RainBO LIFE15 CCA/IT/000035 (Alberoni et al., 2018). Received powers are measured by the provider with the resolution of $1dB$ at a frequency of ten times per second for maintenance purposes, but only maximum ($P_{max}$) and minimum ($P_{min}$) readings in a time window of 15 minutes are stored for backup. Therefore, data is in the format of 15 minutes $[P_{min}, P_{max}]$ pairs. All the available 357 CMLs are "duplex" links, so that two sub-links (back and forth) are present for the same link (although not always simultaneously active). Signal polarization is vertical for 259 CMLs, horizontal for the remaining 98, while carrier-signal frequencies span from 6 to 42.6 GHz, with an average frequency ($\overline{f}$) of 22.1 GHz. Sublinks of the same CML always share the same polarization and differ only in frequency by a small gap of around 1 GHz. Path lengths of the links vary from 162 m to 30 km, the interquartile range extends between 2.4 and 8 km, and the average length is 6 km. As expected, the carrier frequency is anti-correlated with path length since high frequencies, while allowing a wider transmission band, are more prone to attenuation compared to the lower frequencies (Leijnse et al., 2008).

### 2.1.1 Coverage and data quality

The number of working CMLs varies slightly over the months: it grows from 348 at the beginning of May to the maximum of 357 in June. The number of valid CMLs for rain retrieval is lower because of the quality and sensitivity filtering performed by the pre-processor of the algorithm (see Section 3.1), resulting in a median number of 308 valid CMLs with very small fluctuations. Most of the rejected data is empty or incomplete ($P_{min}$ or $P_{max}$ missing), probably due to failures in reading or storing raw data. More details on the rejected data are presented in the Supplement.

Four parameters are utilized to summarize the topological structure of the CML network: the link density $LD$ (defined as the total number of link paths divided by the whole area, in $km^{-2}$), the average link length $LL$ (in km), the bulk link coverage $BC$ (defined as the sum of the link path lengths divided by the total area, in $km\,km^{-2}$) and the local link coverage $LC$ (calculated as $BC$ but for each gridbox, in $km\,km^{-2}$). Due to Vodafone confidentiality restrictions, we are not allowed to show the exact location of the available links, so we show, instead, in Figure 1 the spatial distribution of $LC$. The regions with the greatest coverage are located where the anthropic presence is highest, i.e. around the two capital cities and along the main communication routes. The hilly region of the province of Parma and the rural plains of the province of Bologna have instead the least covered grid boxes.

Since the RAINLINK original settings depend on the network characteristics, we compared the Emilia-Romagna network (ER) with the one from The Netherlands (NL), which is included in the RAINLINK software package as test sample (Overeem et al., 2016a), and with other datasets on which the algorithm was employed (Overeem et al., 2013, 2016b). The datasets properties are summarized and compared in Table 1. ER has comparable link density and higher average link length, resulting

in a higher bulk coverage with respect to the NL network. The province of Bologna hosts more than half of the links (195 against 113) and thus has a higher $LD$.

### 2.1.2 Transmitting power levels

CMLs are usually equipped with Automatic Transmit Power Control devices (ATPC) which modulate the transmit power to guarantee a constant power level at the receiving end of the link, cancelling minor fluctuations of the total attenuation along the path. ATPC works at a higher frequency than 15 $\min^{-1}$ and in a power window spanning from 0 to +6 dB. With ATPC active, attenuation measurements should, therefore, be performed subtracting receiving to transmitting powers and are not possible from receiving powers only (Overeem et al., 2016a). The CMLs analysed in this work are equipped with ATPC, but
unfortunately we do not have access to the transmitting powers, due to confidentiality restrictions. Luckily, provider engineers gave us instead some ATPC data – specifically, the maximum modulation offsets (in dB) that were applied during each time interval – through which we are able to correct the receiving power levels, compensating for the power modulation effects and simulating CML data with constant transmitting powers, allowing RAINLINK to estimate attenuations from receiving powers only. The correction intervenes only on minimum received powers ($P_{min}$), which are with no doubt affected by the ATPC:
they are manually lowered by the maximum ATPC modulation applied within the respective 15 min time window. Maximum receiving powers ($P_{max}$) instead are left untouched as the ATPC working frequency and the 15 $\min^{-1}$ sampling frequency does not coincide and there was no way to infer a reasonable compensation. This could result in a broader gap between $P_{min}$ and $P_{max}$.

## 2.2 Reference rainrate fields

### 2.2.1 Rain gauges

Rain gauges hourly and 15 min data are provided by Arpae RIRER (regional hydro-meteorological network), established in 2001 by bringing together existing hydrological and meteorological station networks, managed at the time by various public bodies and local authorities. The network of the whole Region is composed of 285 stations, equipped with tipping bucket rain gauges: 110 of them are divided between Bologna (54) and Parma (56) provinces. Rain gauges have different sampling
intervals (from 10 to 60 minutes), they undergo a process of homogenization and quality control and are released as an hourly point-like product.

### 2.2.2 ERG5 rainfall analysis

The ERG5 gridded meteorological data set has been developed by Arpae-SIMC, to support agricultural activities in the region of Emilia-Romagna. ERG5 data are operationally produced since 2001, interpolating the hourly station measurements of the
150 main meteorological variables (air temperature, relative humidity, precipitation, wind, solar irradiance) onto a 5km×5km grid covering the Emilia-Romagna region. The interpolation method used for hourly precipitation consists of a Shepard (1968) modified scheme using topographic distances instead of Cartesian distances. This allows the interpolation to take into account

the influence of topography on precipitation, by making locations separated by orographic obstacles more distant than they would be if Cartesian distances were used (Antolini et al., 2016). Data are stored and distributed freely in the form of GRIB2 files, which were imported in an R environment thanks to the rNOMADS package (Bowman and Lees, 2015). Among all the variables included in ERG5, we consider here only the hourly accumulated precipitation. Its input is based on the same RIRER network described in the previous Section, no longer limited to the two areas of study, but extended to the whole Region. Some discrepancies are therefore expected between the two products, mainly near the borders and in areas where the distribution of the instruments is less uniform.

### 2.2.3 Radars

Radar data set is based on hourly precipitation estimates obtained from the composite of the regional radar network managed by Arpae-SIMC. The regional network is composed of two C-Band systems, located in San Pietro Capofiume and in Gattatico (easternmost and westernmost red crosses in Figure 1, respectively). For each instrument the equivalent radar reflectivity factor close to the ground is extracted and interpolated from polar coordinates to a $256 \times 256$ Cartesian grid of $1\text{km} \times 1\text{km}$ resolution, then merged to obtain a composite of both radars.

Raw radar images are affected by various non-meteorological echoes that are removed before computing the Quantitative Precipitation Estimation (QPE). The current scheme used at Arpae-SIMC during operational service includes many steps: the ground clutter is removed at first statically through the map of signal-free elevations recorded in dry conditions, then dynamically by combining a beam trajectory simulation at the current atmospheric state (as measured by radio soundings) and a Digital Elevation Model (Fornasiero et al., 2006). The beam blocking reduction and correction is performed based on a geometric optic approach (Bech et al., 2003), while anomalous propagation is detected after the analysis of the echo coherence in the vertical direction (Alberoni et al., 2001). The final conversion between reflectivity and rainfall rate is performed on the corrected data set using the classic relationship $Z = aR^b$, with $a = 200$ and $b = 1.6$.

Rain rates are obtained every 5 minutes and hourly total rain amount is computed by an advection algorithm which takes into account the movement of the precipitating systems. The algorithm is based on the computation of maximum cross-correlation between consecutive maps, leading to the estimate of the displacement vector for each precipitating system. The rainfall field is then reconstructed every minute between the observations and cumulated over each hour. Finally, radar QPE is adjusted with rain gauges data, via the spatial analysis of the ratio $G/R$ between rain gauges ($G$) and radar ($R$) rainfall rates over the station locations. The spatial analysis is obtained as the weighted mean of the $G/R$ values where the weight is a function of both the distance of the grid point from the station and the mean spacing between 5 observations (Koistinen and Puhakka, 1981; Amorati et al., 2012). In this work we will compare the CML product with both adjusted and unadjusted radar QPEs.

### 3 Methodology

The process chain which takes CML signals and returns rainfall maps is governed by the RAINLINK algorithm (Overeem et al., 2016a) published open source on GitHub (https://github.com/overeem11/RAINLINK) as an R package. We used the

1.14 version of the RAINLINK algorithm, available online from July 2019, and we added some minor modifications and optimizations (forked version available here: https://github.com/giacom0rovers1/RAINLINK).

## 3.1   CML rain retrieval algorithm

The algorithm works for both instantaneous power measurements and $[P_{min}, P_{max}]$ pairs: for the present work we use the latter, on 15 minutes intervals. The algorithm treats $P_{min}$ and $P_{max}$ separately (we will then use $P_i$ to refer to both alternatively).
Two separate rain estimates $R_{min}$ and $R_{max}$ will thus be obtained. The retrieval process is summarized below, while we show more details of the data filtering in the Supplementary Material.

1. **Preprocessing**: the raw input goes through three consistency checks concerning data formatting and labelling. Any multiple observations for the same LinkID and DateTime are discarded, each LinkID is verified to maintain the same metadata throughout the whole dataset (Frequency, PathLength and antenna coordinates) and rows with NA values in
any of the columns except for Polarization (which is supposed vertical if not indicated) are discarded as well.

2. **Wet-Dry Classification**: the samples are discriminated in wet and dry periods by assuming that rainfall is correlated in space, through the so-called Nearby Links Approach (NLA), which works as follows. For each link, a time interval with a decrease in the received power is labelled as wet if at least half of the links in the vicinity (within 15 km radius) experience a comparable reduction, i.e. if the medians of the attenuation and the specific attenuation of the nearby links
are below $-1.4$ dB and $-0.7$ dBkm$^{-1}$ respectively. This is the second most computationally time-consuming step of the algorithm.

3. **Baseline determination**: a 24 h moving-window median of the quantity $\frac{1}{2}(P_{min} + P_{max})$ over the dry time intervals defines a reference level $P_{ref}$ (baseline). This is the computationally time-consuming operation of the algorithm.

4. **Outliers filter and power correction**: outliers due to malfunctioning links can be removed again by assuming that
rainfall is correlated in space. The filter discards a time interval of a link for which the cumulative difference between its specific attenuation and that of the surrounding links over the previous 24 h (including the current time interval) becomes lower than the outlier filter threshold, which is fixed at -32.5 dBkm$^{-1}$h. After removing the outliers, the classification information is used to clean the receiving powers of the noise over the dry periods. The corrected powers $P_i^{Cor}$ will be equal to $P_{ref}$ on dry periods and $P_i$ on wet ones.

5. **Rainrate retrieval**: attenuation $A_i$ is computed as $A_i = P_{ref} - P_i^{Cor}$. A fixed quantity $Aa = 2.3$ dB is subtracted from the attenuation $A_i$ in order to take into account the wet-antenna effect, which is independent on path length and it is assumed independent also on frequency and rain intensity. If $A_i - Aa > 0$ then the specific attenuation $k_i$ (dB km$^{-1}$) is calculated as $k_i = (A_i - Aa)/L$, otherwise 0 is returned. Path-averaged mean rain intensities $R_i$ (mm h$^{-1}$) are finally calculated using the $k - R$ relationship $R_i = a(k_i)^b$, where the coefficients a and b were from Leinse (2007) and Leijnse
et al. (2010) for vertical and horizontal polarization, respectively.

6. **Path-averaged rainfall depth**: to obtain a single path averaged rain depth, $R_i$ are combined through a weighted mean: $R = \frac{1}{4}\left[\alpha R_{min} + (1-\alpha)R_{max}\right]$. The factor $\frac{1}{4}$ transforms rain rates in 15 min rain depths. The weight $\alpha$ varies between 0 (estimate derived from $P_{max}$ only) and 1 (estimate derived from $P_{min}$ only); we adopted the default value ($\alpha = 0.33$). We specify that, unlike Overeem et al. (2016a), we chose to keep the subscripts related to the original receiving powers, thus in our notation the rainrate $R_{min}$ is higher than $R_{max}$ because it is obtained from the most attenuated signal $P_{min}$.

7. **Interpolation**: CML path averaged precipitation estimates are assigned to the mid points of the links like point measurements ("virtual rain gauges"). Interpolation of the point-like measurements is performed at hourly scale with ordinary kriging on a spherical semivariogram on the ERG5 grid. Sill and Range parameters are estimated from the available rain gauge stations of three consecutive years. Nugget parameter is set as 1/10 of the Sill, as in Overeem et al. (2016a). The interpolated field is truncated if it gets smaller than 0.05 mm, which is half of the minimum detectable rain from a rain gauge.

## 3.2 Preliminary discussion about the RAINLINK set-up for northern Italy

The implementation of RAINLINK in Emilia-Romagna required some technical and conceptual considerations, based on the differences and similarities between the local and Dutch climatology, orography and CML network features. We will describe them below:

– The CMLs' operational frequency in our data set spans between 5.0 and 45.0 GHz. The default frequency allowance window of the RAINLINK algorithm is 12.5 - 40.5 GHz instead. We decided to extend it to 10.0 - 45.0 GHz but no further, so five CMLs belonging to the 5 to 10 GHz interval were left out. We then removed 10 other links, which had higher frequencies, but whose sensitivities were below 0.1 dB per $\text{mm}^{-1}$ (see Supplement for more details). This is done to avoid contamination from coarse, low sensitivity signals.

– The average link density (0.043 $\text{km km}^{-2}$) is the same as the one of the network used for the original setup of the algorithm (see Table 1).

– Spherical variogram parameters (see Section 3.1, point 7. Interpolation) were calculated for three years from a pool of validated raingauges from the entire region. Range and Sill are resp. 36.12 km and 1.12 $\text{mm}^2$. These values resemble very much the median for May and June of the outputs of the "ClimVarParam" sub-function of Overeem et al. (2016a), which approximates 30 years of Dutch climate (van de Beek et al., 2012).

Accordingly, it is expected that both the network structure and the rainfall spatial patterns are similar between the Italian and Dutch sites. This assumption drives the choice for the correct value of the NLA radius of the wet-dry classification algorithm.

– The differences from The Netherlands regarding orography are more relevant (see Section 2). We expect that rainfall patterns could deviate from the average behaviours described by the variograms when interacting with the complex

orography of the hilly part of the region. However, we do not have enough data to calibrate the NLA radius at a small scale or considering geographical sub-samples. Moreover, a shorter NLA radius could theoretically improve the consistency with the expected decorrelation length, but, given the network in the hilly region mostly consists of medium to long links, candidates which will fall inside the NLA radius could be too few to ensure a statistical significance of the samples. Thus, we left its value unaltered at 15 km, but we are expecting that some issues could possibly arise in the areas characterized by the most heterogeneous terrain.

– The default k-R relationship from Overeem et al. (2016a) is also maintained as is, since northern Italy and The Netherlands share a similar climate: the average drop size distributions (DSD) differences between the two countries are expected to be negligible (Caracciolo et al., 2006) and certainly lower than the expected variations of DSD along the link paths and during the 15 minutes time intervals (Tokay et al., 2017).

All the other algorithm's parameters were not specifically calibrated. The reasons behind this out-of-the box approach are numerous:

– As suggested by their authors (Overeem et al., 2016a), a solid calibration of the RAINLINK retrieval algorithm should be implemented exploiting numerous instruments along the link paths and organizing dedicated measurement campaigns, which were not feasible for us.

– The overall temporal span should also allow the dataset to be split into two non-overlapping data sets for calibration and validation, but the total wet hours available to us were not enough to grant a statistical significance for both sub sets.

– The gauge-adjusted radar product (which is commonly exploited in most CML studies) is not the one currently selected by the regional weather agency Arpae-SIMC as their quantitative reference, a choice that went in favour of the interpolated rain gauges product ERG5 (see Section 2.2.2). The spatial and temporal resolution of ERG5, however, is too low to perform an effective calibration.

Therefore we analysed some CML - rain gauge pairs only where the gauges were already in the vicinity of the links (Section 4.1), while we validated the rest of the dataset against the reference only through its interpolated product (Section 4.2.1 and 4.2.3).

We consider RAINLINK's ability to function as a standalone system – while other approaches rely on gauges or radars for Wet-Dry classification – as one of its key features. However, since RAINLINK does not include any standardised algorithm or procedure for calibration, performing it would lead to a huge increase in the set up efforts, which would make other algorithms (where adaptation to local characteristics is naturally present, e.g. neural networks) much more competitive.

## 3.3 Error metrics

In the present work, we selected two sets of classical skill indicators, broadly used in the validation community (Nurmi, 2003): the first one is to assess the capability of the product to detect rainfall occurrence (categorical indicators), and the second one is

to evaluate the skill in correctly estimating the quantitative precipitation rate (continuous indicators). The first set is computed after a definition of a confusion matrix by counting the number of samples where both estimate and observation agree on classifying wet (hits, H), or dry (correct negatives, CN) samples, and where there are misses (M, observed wet and estimated dry) or false alarms (F, observed dry and estimated wet). Namely, Probability of Detection, False Alarm Ratio, Multiplicative Bias and Equitable Threat Score are defined resp. as:

$$POD = \frac{H}{H + M} \tag{1}$$

$$FAR = \frac{F}{H + F} \tag{2}$$

$$MB = \frac{H + F}{H + M} \tag{3}$$

$$ETS = \frac{H - H_{rnd}}{H + M + F} \tag{4}$$

where $H_{rnd}$ represents the number of hits obtained by chance.

Given $e_i$ and $o_i$ as estimated and observed values respectively, continuous indicators are the normalized Mean Error and the normalized Mean Absolute Error, defined as:

$$ME = \frac{\sum_i (e_i - o_i)}{\bar{o}} \tag{5}$$

$$MAE = \frac{\sum_i \|e_i - o_i\|}{\bar{o}} \tag{6}$$

plus the Coefficient of Variation (CV), defined as the root mean square error divided by the mean of the observed values $\bar{o}$, and the Pearsons' Correlation Coefficient (CC), as the covariance of observed $o_i$ and estimated values $e_i$ divided by the product of the two standard deviations (Nurmi, 2003; Overeem et al., 2016b).

Both interpolated CML and reference field have a large number of very low positive values (below 0.1 mm h$^{-1}$) that do not have any physical relevance, but which are potentially very influential in normalized error metrics. Thus we have set a wet-dry threshold equal to the minimum rain quantity detected by the tipping bucket rain gauge, i.e. 0.1 mm h$^{-1}$, for both estimate and reference. Categorical indicators are calculated with respect to this threshold for the whole dataset, while all the continuous indicators are computed only for the product-reference pairs where both values exceed the threshold (i.e. wet-wet). ME, MAE and CV are normalized with the averaged reference rain depth.

## 4  Comparison between CML and conventional precipitation products

We carried out the validation of CML product at three different levels. First, we compared single link estimates with the measurements of a nearby rain gauge, at the shortest temporal scale available (15 minutes), to discuss success and failure cases, trying to understand the latter. Secondly, we compared the interpolated 5km×5km CML hourly rainfall maps versus the ERG5 product at grid box scale, also analysing three case studies. In the third step, the map comparison is carried out at a basin scale including even the other precipitation products available at Arpae-SIMC.

## 4.1 Single link verification

We have selected links in rural areas and different terrains with an active rain gauge close to the link: the distance between link and rain gauge, reported in Figure 2, is always below 3 km (significantly lower than the correlation distance of precipitation in Italy (Puca et al., 2014)) and always lower than the length of the link itself. In general, no dependence of the link performance on the distance from the rain gauge is found. Selected links had to be active for all the analysed period. In many cases more than one link was selected for one rain gauge. Temporal sampling is kept at the highest frequency, which is a measurement every 15 minutes for both the CML and the rain gauges. 12 rain gauges and 26 CMLs have been chosen, 14 of which are in the northern part of the domain and the other 12 on the hilly region at elevations between 193 m and 960 m a.s.l..

The rain depths of the 26 CMLs are reported in Figure 2 for the whole study period, grouped accordingly to the closest rain gauge and ranked by its altitude. A large variability is found (ranging from near-perfect agreement to discrepancy of a factor of 2 or 3 in the worst cases). 75% of the 26 links' CCs are between 0.5 and 0.88, with overall median value 0.68, proving an acceptable overall skill. We relate this variability to the heterogeneity of CML sensitivity, the small scale of the meteorological events (see Supplement) and different site exposure and elevation. In most cases, CMLs underestimate the rain gauge values: the links located in the lowlands (Figure 2a, 2b, 2d and 2e) show a better correspondence than those in the hilly regions, where underestimation is more significant.

In some cases (Figure 2f, 2k and 2l) the discrepancies between CMLs close to the same rain gauge (but different in location, frequency and length) are much lower than the CML-rain gauge differences: all these CMLs are in good mutual agreement and share the same classification issues, resulting in a systematic underestimation which therefore seems to be caused by the algorithm setup. In other cases (Figure 2b, 2d and 2g) some links clearly outperform other members of the same group. This second kind of discrepancies is more likely related to real differences, like inhomogeneous rainy structures which crossed the link paths or different hardware setups, while there is no evidence of a correlation with frequency or path length. The difference between the two directions of the same link is generally below 10%, except for the Ostia Parmense site (see Figure 2g).

To gain a deeper understanding of better and worse performance of the single links, we performed a more detailed analysis of case studies at the rain-event scale (Figure 3). We show a case when the link retrievals accurately match the measurements of the close-by rain gauge, and a case with markedly low performance. In Figure 3, graph panels are organized in columns by CML and in rows by sub-link. In the top panels are shown all the signals managed by the algorithm: the reference power $P_{ref}$, the raw received powers $P_{min}$ and $P_{max}$ and the filtered received powers $P_{min}^{Cor}$ and $P_{max}^{Cor}$. In the middle panel rain gauge measurements are compared with CML estimates and also the minimum and maximum attenuation signals are plotted ($A_{max}$ and $A_{min}$ respectively). The grey background indicates when the classification detects a dry period. The pink background indicates the band inside which attenuation is considered as caused by a wet antenna ($Aa$ parameter) and is discarded for rain retrieval. The bottom panels show the cumulated rainfall depths in the same time frame.

### 4.1.1 Best cases example

Between 11 and 12 May 2016 an extensive convective system covered the Bologna Province area almost entirely, with a
340 maximum rainrate of 23 mm h$^{-1}$, and widespread precipitation around. For this case, the NLA classification on the three
links near Sant'Agata (Bologna Province, 18 m a.s.l.) works properly: in Figure 3b most of the measured rain is on white
background. In Figure 3a, after the attenuation event, the noisy signal is correctly filtered, and a very small amount of rain
(just above the gauge threshold) is neglected. The agreement is qualitatively very high between each pair of sub-links and
good among the different links, in terms of specific attenuations and retrieved quantities (see Figure 3b). Quantitative retrievals
give some overestimation for one of the CMLs, whose effect is evident on the accumulation plot (Figure 3c) where the total
rain depths are compared. During the two months the Sant'Agata links are generally in good agreement with the close-by rain
gauge, with CC ranging between 0.66 and 0.88 and CV between 0.47 and 0.96.

### 4.1.2 Worst cases example

Between 8 and 10 June 2016 an event hit the Vergato site (Bologna Province, 193 m a.s.l.). It was characterized by intense
rainfall peaks (rainrate up to 14.6 mm h$^{-1}$) and iterated moderate scattered precipitation. Many wet intervals are missed due
to wet-dry misclassification (Figure 3e), leading to a 20 mm loss in the rain accumulation (Figure 3f). The POD over the entire
period for these two links is between 0.22 and 0.29.

In the case when the NLA classification correctly identifies some rain occurrence, there is still a general quantitative under-
estimation. It could be seen that half of the signal is hidden from the wet antenna attenuation threshold. The continuous scores
for the wet-wet sample on the entire period show a good correlation with gauges but are poor in statistical relevance because
of the high number of misses. They nevertheless confirm the tendency to underestimate, by around 40% (ME=-0.40).

### 4.2 Gridded product verification

The verification of the RAINLINK gridded product (1 h cumulated on the 5km×5km grid) with respect to the ERG5 product
is first performed at the highest available resolution (grid box by grid box), since the two products intentionally share the same
interpolation grid. Secondly, the comparison is carried out at the basin scale by matching spatially averaged time series over
areas of different size, in parallel with other operational precipitation products available at Arpae-SIMC.

### 4.2.1 Highest resolution matching

Figure 5 shows a scatter density plot for the whole dataset over the entire period. CML estimates from RAINLINK in northern
Italy over uneven ground have an overall underestimating performance of -26% on the accumulated rain over the two months.
The $CV$ is 0.78 and $R^2$ (the square of the Pearson's correlation coefficient $CC$) is 0.46, based on a sample of 10672 total wet
hours. To make the comparison with past works easier, we computed continuous indicators with the filter set as Reference >
0.1 mm and with no filtering at all. Results with the first setting yield worse indicators, increasing the $ME$ to -0.41 and the
$CV$ to 0.95, with a second digit increase for $R^2$, around 0.5. The no-filter run shows values of $ME = -0.33$ and $R^2 = 0.53$

which are aligned with our most filtered results, while $CV = 4.6$ is greatly affected by very small rainrates. These results are
in agreement with similar studies (Overeem et al., 2013, 2016b) despite the differences in the products involved: comparison
between our results, with both filters, and the ones presented in the mentioned works are shown in Table 2.

The performance of the rain detection capabilities with respect to the $0.1\,\mathrm{mm}$ threshold is evaluated by the set of categorical
scores defined in Section 3.3. Quantitative continuous indicators from now on are computed only for the grid boxes where both
CML and ERG5 reported more than $0.1\,\mathrm{mm}$ at the same time. Categorical and continuous indicators are evaluated for five
areas, with a different extension (S) and average Link Coverage ($\overline{LC}$). They are reported in Table 3, ranked according to the
$\overline{LC}$ value: Parma Province (PP), Total Area (TA), Parma River Basin (PRB), Bologna Province (BP), Reno River Basin (RRB).
The total area and the two provinces do not have any specific hydrological meaning, but could be seen as a good foretype of
larger river basins with heterogeneous terrain (see Figure 1). All normalized indicators are relative to the average reference
(ERG5) rain rate. Numbers in bold (italics) are the best (worst) value in the column.

We found ETS values ranging from 0.38 to 0.43, which are comparable with the ones obtained from satellite observations
(Puca et al., 2014; Feidas et al., 2018) in similar regions. For four out of five areas (excluding RRB for now) the RAINLINK
product underestimates the rain occurrence (MB < 1), with a relatively low value of POD (0.48 to 0.57). The FAR is also rather
small, (0.28 to 0.32), resulting in ETS values (0.38 to 0.43). Mean Error confirms the underestimation of rain amount (ME
between -0.18 and -0.34), CV ranges between 0.73 and 0.80, CC between 0.62 and 0.74. For comparison, Petracca et al. (2018)
analysed over Italy the instantaneous estimate of the Global Precipitation Measurement - Dual-frequency Precipitation Radar
(GPM-DPR), considered as the most reliable and accurate instrument to measure precipitation from space. Over a footprint
of a size comparable to the one used in this paper, the best value of CC is 0.57, while the CV was between 1 and 2. Other
validation studies of GPM-DPR products in the alpine region (Speirs et al., 2017) obtained relatively good POD (up to 0.78),
FAR (below 0.08) and CC (up to 0.63) over flat terrain, with a dramatic drop of the skill indicators when areas with complex
topography are considered.

The averages over the Reno River Basin stand out for all the indicators, either positively or negatively; therefore they need a
separate description. As highlighted in Table 3 in bold and italics fonts, RRB has half the FAR the other samples have (0.16),
almost ten points less CV (0.62) and nearly fifteen points better CC (0.8, which is unexpectedly high), with the mean errors
aligned to the other samples. The higher accuracy in the estimates is reached at the expense of POD, ETS and MB: around
50% of the rainfall duration is lost in this area. The main peculiarity of the RRB area is the high $\overline{LC}$, which is 50% higher than
the rest of the regions.

The marked improvement of continuous indicators for RRB suggests that the quantitative matching between estimated and
reference could be positively related to $\overline{LC}$. Thus, we further investigate its effect on scores by grouping each grid box by
LC quartiles, regardless of the actual geographical location, and reported the results in Figure 6. Five out of six indicators
improve as LC increases (FAR, MAE, ETS, CC and CV), among which the most striking is the FAR, while POD remains
mostly unchanged, allowing the ETS improvement.

### 4.2.2 Case studies

To assess the performance of RAINLINK with respect to the structure of rainfall fields we focused the analysis on three
one-day-long events with different characteristics, for which RAINLINK provided results of varying quality.

The best performance was achieved on May 19 (see Figure 4, left), when an intense event was characterized by a few
convective episodes on the Apennines, in the Parma Province. Precipitation peaks were around 90 $\mathrm{mm\ day^{-1}}$ (see Figure 4c),
maximum and mean hourly rainrates were about 24 and 2.6 $\mathrm{mm\ h^{-1}}$, respectively (see Table 4). A large area of widespread
moderate precipitation over the Bologna Province (Figure 4a) is also present. RAINLINK is able to localize precipitation local
maxima (Fig. 4b), even if it occurred in areas where link coverage is relatively poor (see Figure 1), providing also accuracy
in the peaks intensity. Estimated PDF matches closely the ERG5 curve, indicating that all rainrates are represented in the
estimates (see Figure 4d). Though, underestimation is present at all ranges, more markedly at the highest rainrates. Numerical
indicators confirm the goodness of the estimate, in terms of wet area detection (ETS=0.59) and relative error (CV=0.69), while
the fractional amount of rain lost by the estimate is low (ME=-0.29).

The second case (11 May) shows a more patchy rainfall field (Figure 4e), which resulted from a series of storms that occurred
in the area during the day. Maximum and mean rates are lower with respect to the first case (Figures 4g, 4h), as well as the
wet fraction of overall samples (see Table 4). Some local peaks are correctly located (especially inside the Bologna Province),
as shown in Figure 4f, and some others, in Parma Province and particularly on the Apennines, are missing. In this case the
underestimation is marked for all rainrates, resulting in higher ME (-0.40) and lower POD (0.66).

A completely different scenario is represented by case three (May 12), when ERG5 measured light to moderate precipitation
(see Figure 4i), with maxima on the Apennines, and a much lower fraction of wet samples. RAINLINK (Figure 4j) is not able
to estimate the highest rainrates, neither to locate the area with the highest intensity. Moreover, it find a spurious peak in the
northern area of the Bologna Province, which is not detected by ERG5. Here the fractional amount of rain loss is -65%, the
POD is low, and an increase of FAR is also to be remarked, indicating that underestimation again dominates throughout the
whole range of rainrates (see Figures 4k, 4l), but in case of light rain, overestimation could also take place.

### 4.2.3 Areal averages matching

In this Section, the matching between estimate and reference field is performed at basin (and Province) scales, comparing
hourly rain amounts averaged over areas of different sizes. The areas selected for this evaluation are the ones introduced in the
previous Section: two of them are chosen because of direct hydrological interest (RRB and PRB), while the other three (BP,
PP and TA) are selected to assess the impact of the increasing target area.

In Table 5 we present the categorical indicators calculated around the 0.1 $\mathrm{mm\ h^{-1}}$ threshold and the continuous indicators
calculated on wet-wet occurrences only, for the five mentioned areas listed this time in order of increasing area size. In general,
best performance is found for the largest areas (BP and TA), while the smallest ones (PRB and RRB) show the worst values.
CML product underestimates precipitation occurrence (MB between 0.41 and 0.70) and amount (ME between -0.18 and -

435 0.34) at all scales. Due to the areal averaging, CC is markedly higher than the high-resolution values reported in Table 3. The characteristic behaviour of RRB (lowest FAR and POD, highest CC) also remains in this case.

The same areal-averaged statistical indicators have also been computed for all the operational products available at Arpae-SIMC for routine use and described in Section 2.2, reported to an hourly scale and compared with the ERG5 product. We show in Figure 7 the values of the statistical indicators as a function of the target area.

The rain gauge product, obtained by averaging the measurements of the rain gauges in the area, performs similarly to its interpolated version ERG5, as expected, and diverges only for small areas, where the impact of a single sensor in disagreement with neighbours is the highest.

Radar product shows, in this metric, almost the same performance both with and without the gauge adjustment[1]. Both have very good detection capabilities (POD is almost 1) but high rates of false alarms (FAR around 0.5) and marked quantitative

discrepancies (MAE around 0.9, CV between 0.75 and 2).

The CML product outperforms both radar products in terms of CC, CV, MAE and FAR, while it lacks in detection capability (CMLs POD between 0.4 and 0.6). Figure 8 shows that the overestimating and underestimating behaviours, of radar and CML products respectively, can be seen as complementary. For radars, the spread is more relevant than for CML, but it has to be remarked that the latter has a smaller sample size due to the already mentioned low POD issues. It has also to be said that part of

450 the radar's high FAR and overestimation could represent real rain from small precipitating structures, often observed between meteorological spring and summer in Italy (see Supplement), that are randomly missed by the rain gauges (and therefore by the ERG5 reference product as well).

In Table 6 the latency and sampling characteristics of the four precipitation products we took for comparison are reported, along with CML product. CMLs operational specifications refer to an implementation of the RAINLINK algorithm as part of

455 a real-time service, tested in 2019 by MEEO S.r.l. within the RainBO project (LIFE15 CCA/IT/000035).

## 5 Discussion

The underestimating behaviour that emerged in the single-link-vs-gauge analysis (Section 4.1) seems to be largely imputable to a wrong wet-dry classification. Though we do not have a data set large enough to support general statements, looking at Figure 3d we could gain some insights about what goes wrong in two of the most problematic CMLs of our population, the

460 Vergato ones.

Most of the rain which is sensed by the gauge is falling in intervals that the NLA reports as "Dry" (grey background). $P_{min}$ in fact clearly experiences some decrease, which is coupled to the missed rainfall, but $P_{max}$ does not. This behaviour of $P_{max}$ is not an issue in itself, as the NLA classification relies on $P_{min}$ only. It indicates, however, that there are power fluctuations which happen faster than 15 $\mathrm{min}^{-1}$, otherwise $P_{max}$ would have decreased too. Rapid fluctuations, in turn, suggest irregular,

---

[1]This is to be expected since the radar adjustment acts only at the rain gauges' locations and does not guarantee the consistency of the areal average of the entire rain field. Furthermore, the adjustment mainly affects rainfall rates higher than our threshold of 0.1 mm and has lower performance as spatial variance increases, e.g. in case of small scale convection.

rapidly varying or scattered precipitation patterns. These are actually elements that could affect the correct classification, since NLA relies on the spatial correlation of the rainfall field in a range of 15 km (see Section 3.2). Therefore, a $P_{max}$ signal which remains always near the baseline could be a precursor of local NLA issues. Classification errors are likely the best explanation for the low POD scores.

Given how we filtered the data (Product > 0.1 and Reference > 0.1), we need a source of error other than the misclassification as responsible for the quantitative underestimation measured by the set of continuous indicators (see Section 3.3 for reference). We saw that half of the signal in the correctly classified interval (Figure 3e, white background) remains under the wet antenna attenuation threshold (pink horizontal band). We can presume that the antenna is actually dry, so the $Aa$ threshold in this case is reasonably too high (as also noted by de Vos et al. (2019)). However, a simple sensitivity test, carried out to assess the impact of a decrement in the $Aa$ threshold on the single-link-vs-gauge scores, did not lead to any substantial improvement, especially if the new value is used to process the whole dataset. More information is provided in the Supplementary Material.

Comparing the average performance of the interpolated product (Section 4.2) above the different sub-areas, particularly with respect to the RRB one, we can infer that higher mean coverage ($\overline{LC}$) leads to a more selective NLA classification, which reduced FAR and POD. When grouping the single grid boxes based on their coverage (see Figure 6), it seems however that the sensitivity to LC could explain only the improvement in FAR, but not the sharp decline in POD, suggesting that LC was probably not the only variable at play in the Reno basin. These results integrate the findings of Overeem et al. (2016b), that highlighted the positive impact of higher $\overline{LC}$ on CV and CC at lower spatial resolution. Other studies will be conducted in the future to gain more insights on these topics.

Looking at a daily scale (Section 4.2.2), the interpolated output of RAINLINK is undoubtedly able to resolve small size, short-living events, and even providing quantitatively accurate estimates. In case of widespread, moderate precipitation the overall rain pattern is still effectively represented, but some underestimation of the numerical values appears. When in presence of light and intermittent rainfall, instead, we see the consequences of the issues emerged during the single-link analysis. The rainfall maps in the panels $i$ and $j$ of Fig. 4 reveal that the discontinuity of the link distribution across the borders of the considered areas could be another possible source of discrepancies. We give more insight about this issue examining the maps of the total rainfall accumulation in the Supplement.

Even knowing that the limitations we have just discussed are not negligible, we can still compare the CML interpolated product's performance against the traditional ones, to see whether some overall sensing skill is present or not. We used the areal-averaged hourly rainfall accumulations (see Section 4.2.3) to compare products with different spatial resolutions. The comparison between radar and CML is particularly interesting as they appear to be rather complementary data sources. CML product in this setup clearly lacks the detection capability (POD) of the radar. CML retrieval process however, being based on electromagnetic attenuation instead of back-scattering, does not share the radar's high sensitivity to the drop size distribution (Leijnse et al., 2008). This could make the CML a more robust sensor, in the sense that the same coefficients can be applied

regardless the different types of rain (convective, stratiform, mixed), and the values of the continuous indicators seem to endorse that.

Alongside the considerations on the sensing skills, it is valuable to a forecaster in an operational context also the time which will elapse from the acquisition of primary data (ideally, the occurrence of the event) to the actual delivery of the product ready for use. We referred to it as "latency". It can be seen from Table 6 that the combination of short-latency and high resolution

provided by CMLs is unmatched by all the other products except the raw radar, which although lacks the required quantitative accuracy. It is left to the operators' preference, based on products' error structure, current meteorological conditions, and customers requirements, to make use of the most suitable product or of a combination of them. CMLs are valuably able to widen the range of available options.

## 6 Conclusions

An assessment of the rainfall retrieval capability of CML opportunistic sensors over complex terrain in northern Italy is conducted at different spatial and temporal scales for two months of data. We implemented the open-source RAINLINK algorithm in a new area and context, where no regional CML studies had previously been performed. We evaluated its performance through a complete validation scheme which utilises operational precipitation products as reference, gauging at the same time also the implementation efforts and identifying major strengths and weaknesses to facilitate profitable use of CML products.

First, 26 CMLs are compared with the closest rain gauges at a 15 min scale. Overestimation and underestimation of rain amount are both present, though the latter appears dominant. A marked variability among different links does not prevent to achieve a generally acceptable skill (CC from 0.50 to 0.88). The wet-dry classification approach (NLA, based on the spatial correlation of the rain) and the value of the wet antenna correction (Aa) may produce some misses in both rainfall occurrence and amount, particularly in case of small scale or intermittent episodes. Finally, CMLs located at higher elevation generally

show worse performance.

Interpolated products obtained from 308 links confirm that a non-negligible quantity of rain is missed (normalized Mean Error is -0.26, overall CC is 0.68, and overall CV is 0.78), but also show that the rain retrieval capability is suitable for the operational application, especially if the product is integrated over large areas (CC rises to 0.92). Higher link densities increase the quality of the CML estimates at both gridbox and basin scales, mostly in terms of decreased FAR.

Performance at the daily scale shows enhanced skill in case of heavy precipitations, even in the case of localized episodes. Problems arise instead during light to moderate rainfalls, when the limitations emerged during the single-link analysis become evident. Negative impact on the overall results comes from areas with poor sensor coverage, especially near the borders of the studied areas, but it should be considered that also reference rainfall fields can be affected by shortcomings of the same nature.

Furthermore, when compared to other products currently available for real-time operational exploitation, the RAINLINK

output shows similar or better abilities, especially if low FAR is valued more than high POD and if latency is also taken into account. The integration of a CML-based product into an operational weather service appears worthwhile, even in a plug-in implementation that omits specific local calibration.

*Code and data availability.* CML data were provided by Vodafone Italia S.p.A.. via direct purchase from MEEO S.r.l. and are not publicly available. Gauge data from Emilia-Romagna are freely available at https://simc.arpae.it/dext3r/. Radar reflectivities in near real-time are freely available at https://www.arpae.it/sim/?osservazioni_e_dati/radar, while derived rain products and ERG5 analyses are available upon request at Arpae-SIMC (https://www.arpae.it/sim/). The core algorithm is available (open source) at https://github.com/giacom0rovers1/RAINLINK and was forked from https://github.com/overeem11/RAINLINK on the 26th of August 2019 (RAINLINK version 1.14).

*Author contributions.* GR adapted the RAINLINK code to Italian data, ran the analysis, plotted the data and contributed to the interpretation of the results and to the writing of the manuscript. PPA and AF performed the reference data pre-processing and contributed to data analysis. FP contributed to the design of the validation strategy, to the interpretation of the results and to the writing of the paper.

*Competing interests.* The Author declare that no competing interests are present.

*Acknowledgements.* This work has been partially funded by the Life EU Project RainBO (LIFE15 CCA/IT/000035). The Authors thank Stefania Pasetti and Marco Folegani of MEEO S.r.l. (www.meeo.it) for their support, and are grateful to D. Vecchiato and A. Viaro of Vodafone Italia S.p.A. for the technical assistance with the data. We also thank Aart Overeem, for having developed and released open source the RAINLINK algorithm and for the kind feedback and support he provided to this research.

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

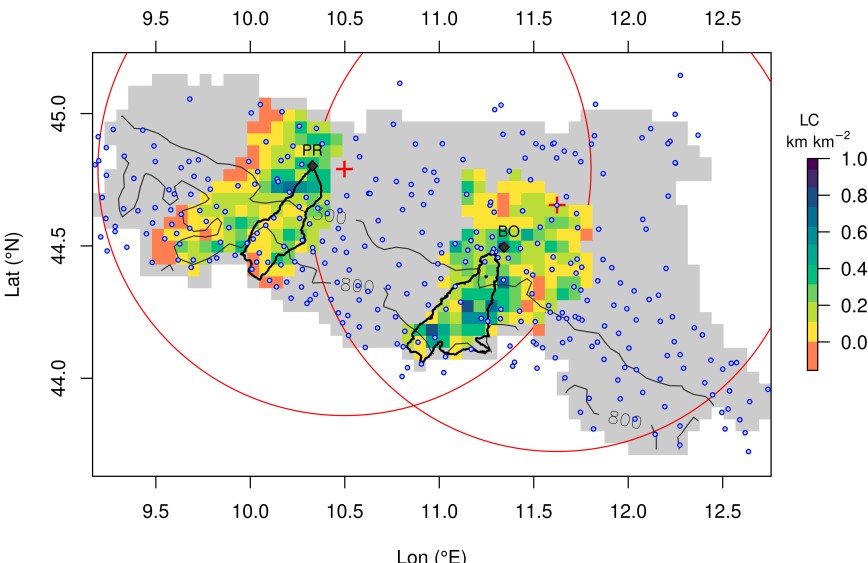

**Figure 1.** Map of the Emilia-Romagna region in Northern Italy (grey area). The coloured areas are the two Provinces where the CML estimates are computed (the colour scale represents the Link Coverage, $LC$, where orange is not a negative value but exactly zero) and black thick lines delimit the two river basins (Parma, to the east, and Reno). Blue dots and red crosses indicate operational raingauge and weather radar locations, respectively, while red circles are the 100 km radar coverage. Thin black lines show two elevation contours (300 and 800 m a.s.l.). The capital cities of the two areas (Bologna and Parma, resp. BO and PR) are indicated with the black diamonds.

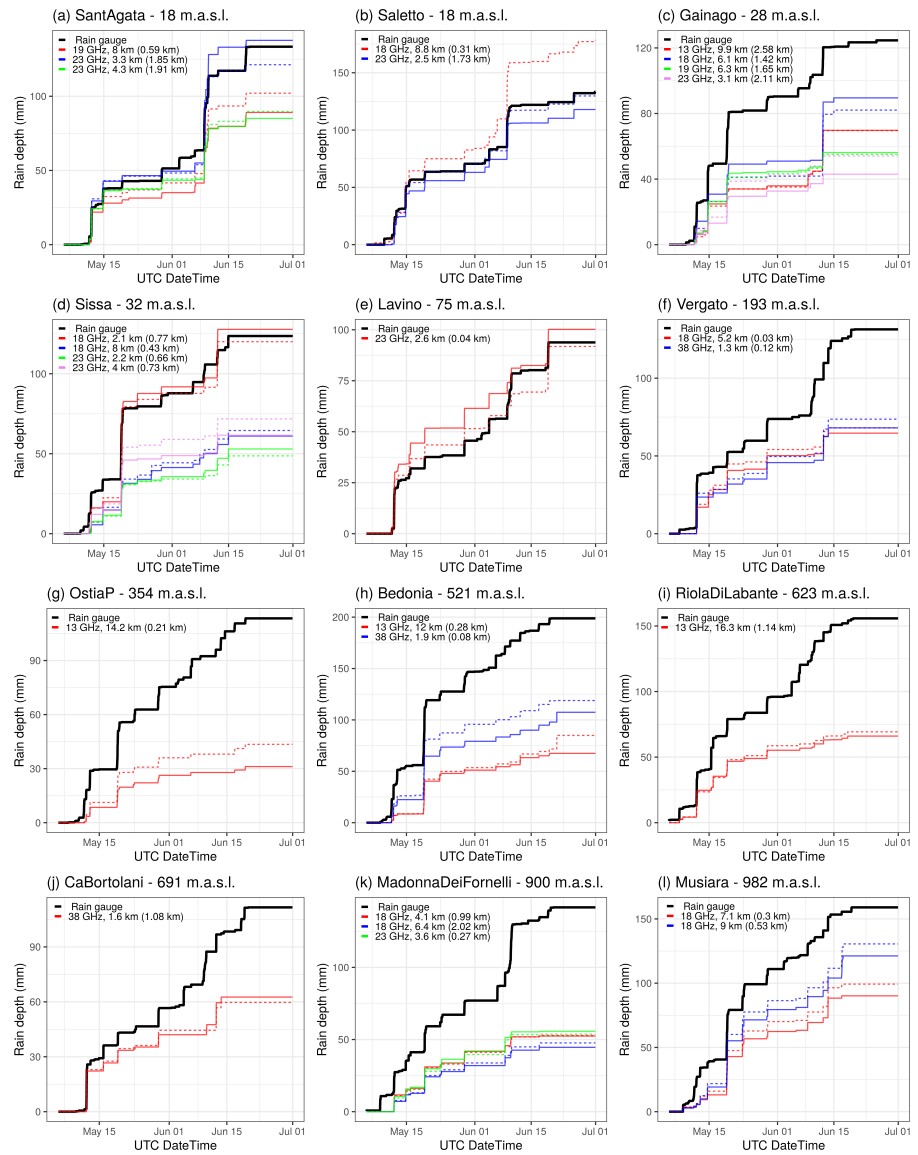

**Figure 2.** Accumulated rain depths over the entire period for the 26 CMLs selected for the single-link analysis. Each tile is named by the corresponding rain gauge, whose accumulated rain depth is shown by the black thick line. Solid and dashed lines represent the two directions (if both active) for every CML (distinguished by colour).

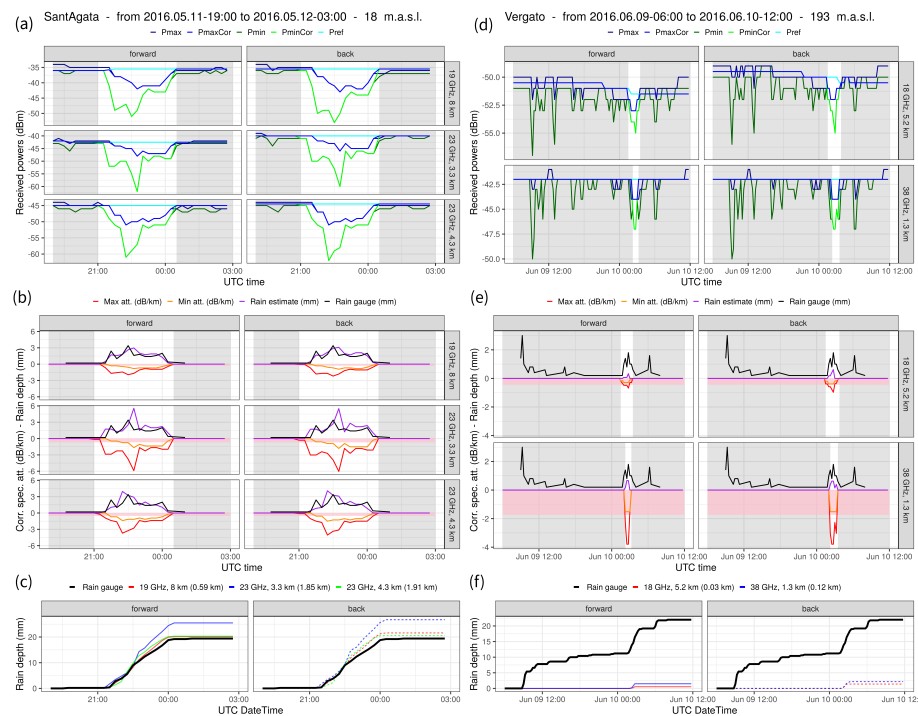

**Figure 3.** Single link analysis for Sant'Agata (from 11.05-19:00UTC to 12.05-03:00UTC) and Vergato (from 09.06-06:00UTC to 10.06-12:00UTC): (a) and (d) show the received signals ($P_{max}$, blue; $P_{max}^{Cor}$, light blue; $P_{min}$, green; $P_{min}^{Cor}$, light green; $P_{ref}$, cyan); (b) and (e) show maximum attenuations (red), minimum attenuation (orange), estimated rainrate (purple), and gauge measurements (black); in (c) and (f) the cumulated raingauge rainrate (black) is plotted with the link estimates. Grey vertical bands correspond to intervals labelled as dry by the NLA classification, pink horizontal bands correspond to the threshold in $\mathrm{dB\,km^{-1}}$ of the Wet Antenna correction of 2.3 dB. Y-axes ranges are specific for each CML as received powers differ between different path lengths.

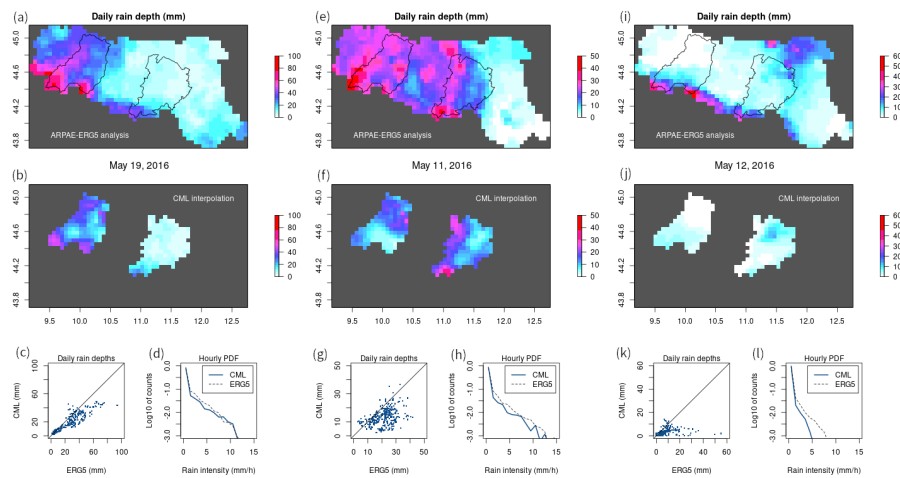

**Figure 4.** Analysis three one-day case studies (May 19, left; May 11, center; May 12, right): (a), (e) and (i) daily cumulated ERG5 precipitation; (b), (f) and (j) daily cumulated RAINLINK precipitation; (c), (g) and (k) scatterplot between the two daily precipitation; (d), (h) and (l) PDF of hourly rain rates.

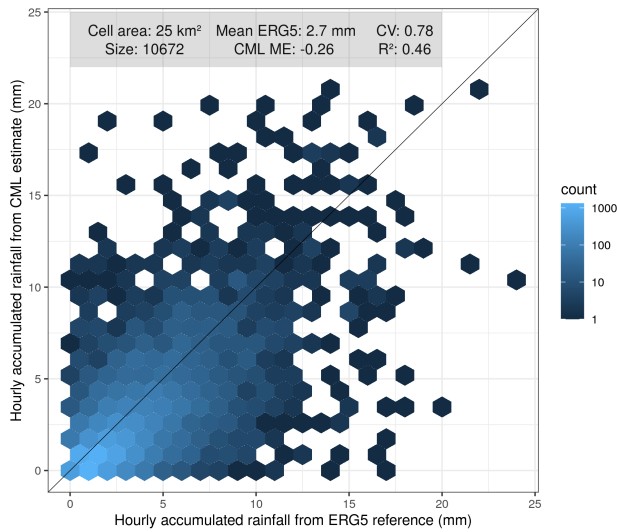

**Figure 5.** Hourly validation of link rainfall maps against ERG5 rainfall maps at grid box scale (highest resolution). Only the rainfall depths in which both CMLs and ERG5 measured > 0.1 mm were used. The black line is the y=x line.

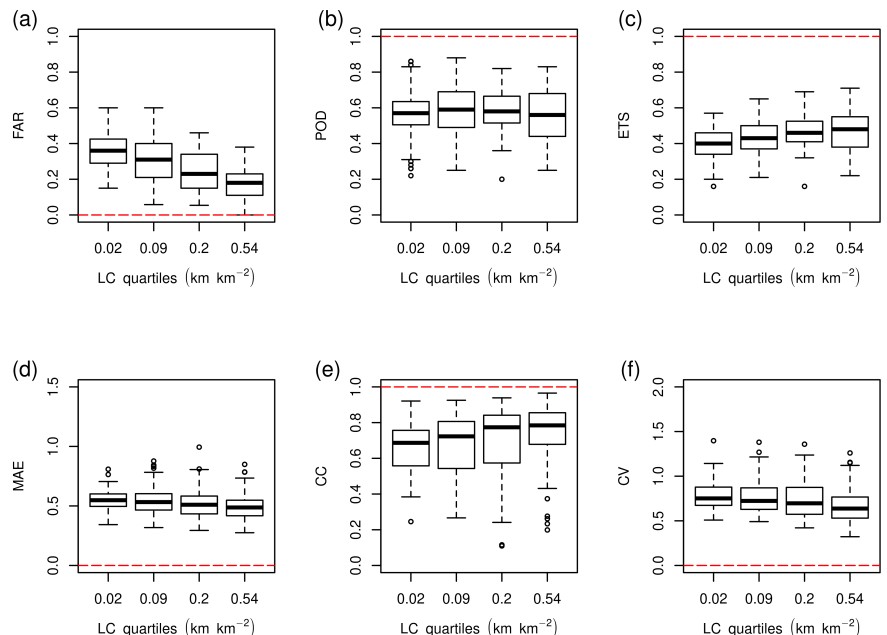

**Figure 6.** Distributions of four statistical indicators computed for every grid box and grouped in boxplots by quartiles of the link coverage $LC$ (labelled by the quartiles centre). Red dashed lines are the optimal values for each score.

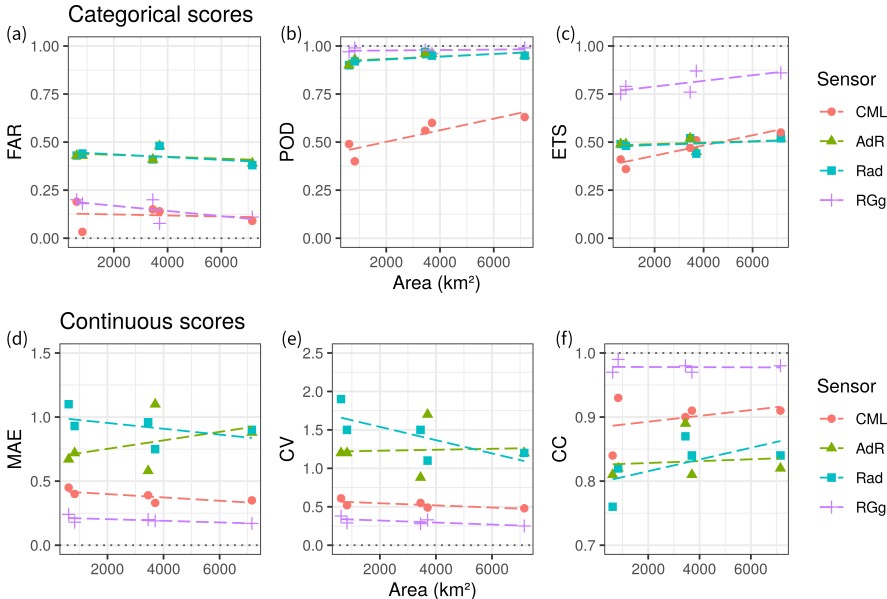

**Figure 7.** Scores of the areal-averaged rainfall amounts grouped per sensor and plotted against basin area. Linear fits are highlighted with dashed lines. The CML scores are also indicated numerically in Table 5.

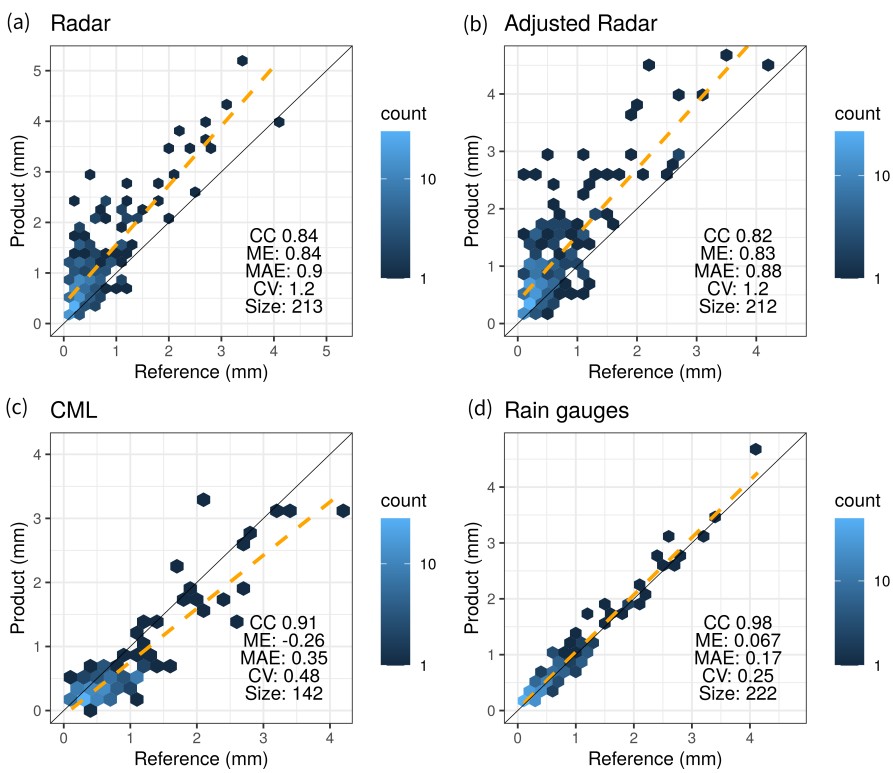

**Figure 8.** Comparison of hourly areal-averaged rainfall depths from the four products against the ERG5 reference. The total area ($TA$) wet-wet hours are considered.

**Table 1.** CML datasets comparison.

| Variable | Unit | ER | NL | Overeem et al. (2013) | Overeem et al. (2016b) |
|---|---|---|---|---|---|
| Total area | $km^2$ | 7149 | 35500 | 35500 | 35500 |
| CMLs | counts | 308 | 1527 | 1514 | 2044 |
| sub-links | counts | 606 | 2473 | 2902 | 3383 |
| LD | $km^2$ | 0.043 | 0.043 | 0.043 | 0.058 |
| LL | km | 5.8 | 2.9 | 3.1 | 3.6 |
| BC | $km\,km^{-2}$ | 0.25 | 0.13 | 0.13 | 0.21 |
| $\overline{f}$ | GHz | 22.1 | 37.11 | 37-40 | 37-40 |

**Table 2.** Comparison with previous studies. Ref isa the reference rainrate, Pr is the product

| Variable | Unit | ER (present work) | | | Overeem et al. (2013) | Overeem et al. (2016b) |
|---|---|---|---|---|---|---|
| Total time window | - | 2 months | | | 3 months | 2.5 years |
| Time scale | min | 60 | | | 15 | 60 |
| Gridbox area | km$^2$ | 25 | | | 81 | 74 |
| Reference | - | interpolated rain gauges | | | Gauge-adjusted radar | Gauge-adjusted radar |
| Filter | - | Ref.$and$.Pr > 0.1 mm | Ref > 0.1 mm | none | Ref.$or$.Pr > 0.1 mm | Ref > 0.1 mm |
| ME | - | -0.26 | -0.41 | -0.33 | 0.02 | -0.16 |
| CV | - | 0.77 | 0.95 | 4.6 | 1.13 | 0.64 |
| R$^2$ | - | 0.47 | 0.50 | 0.53 | 0.49 | 0.49 |

numbers in bold are obtained by performing the validation with the filter Ref. > 0.1 mmmm only

**Table 3.** Statistical indicators for each considered area, considering the highest resolution information (grid box scale), shown in ascending order of $\overline{LC}$. Continuous indicators are normalized and fractional. Values in bold (italics) are the best (worst) values in the column.

| Area | $\overline{LC}$ (km km$^{-2}$) | $S$ (km$^2$) | FAR | POD | ETS | MB | ME | MAE | CV | CC |
|------|------|------|------|------|------|------|------|------|------|------|
| PP | 0.17 | 3447 | 0.28 | 0.51 | 0.41 | 0.71 | *-0.34* | *0.55* | *0.80* | *0.62* |
| TA | 0.18 | 7149 | 0.30 | 0.54 | 0.42 | 0.77 | -0.26 | 0.52 | 0.77 | 0.68 |
| PRB | 0.19 | 624 | 0.30 | 0.48 | 0.38 | 0.69 | -0.31 | 0.50 | 0.76 | 0.67 |
| BP | 0.19 | 3702 | *0.32* | **0.57** | **0.43** | **0.83** | **-0.18** | 0.48 | 0.73 | 0.74 |
| RRB | 0.29 | 828 | **0.16** | *0.39* | *0.35* | *0.47* | -0.31 | **0.45** | **0.62** | **0.80** |

**Table 4.** Rainfall characteristics and performance indicators for the three one-day case studies

| Date | mean R (mm) | max R (mm) | wet fraction | FAR | POD | ETS | ME | CV | CC |
|------|-------------|------------|--------------|------|------|------|-------|------|------|
| 19.05 | 2.60 | 24.0 | 0.37 | 0.10 | 0.77 | 0.59 | -0.29 | 0.69 | 0.78 |
| 11.05 | 2.50 | 21.0 | 0.35 | 0.10 | 0.66 | 0.49 | -0.40 | 0.76 | 0.82 |
| 12.05 | 1.80 | 14.0 | 0.16 | 0.20 | 0.58 | 0.46 | -0.65 | 1.10 | 0.46 |

**Table 5.** Values of the statistical indicators for the mean rain amounts over each considered area, shown in ascending order of surface area $S$. Values in bold (italics) are the best (worst) values in the column.

| Area | $S$ (km$^2$) | $\overline{LC}$ (km km$^{-2}$) | FAR | POD | ETS | MB | ME | MAE | CV | CC |
|------|------|------|------|------|------|------|------|------|------|------|
| PRB | 624 | 0.19 | *0.18* | 0.51 | 0.43 | 0.63 | -0.34 | *0.45* | *0.61* | *0.84* |
| RRB | 828 | 0.29 | **0.03** | *0.40* | *0.36* | *0.41* | *-0.34* | 0.40 | 0.52 | **0.93** |
| PP | 3447 | 0.17 | 0.14 | 0.57 | 0.48 | 0.66 | -0.34 | 0.48 | 0.56 | 0.98 |
| BP | 3702 | 0.19 | 0.14 | 0.60 | 0.51 | **0.70** | **-0.18** | **0.33** | 0.49 | 0.91 |
| TA | 7149 | 0.18 | 0.10 | **0.64** | **0.55** | **0.70** | -0.26 | 0.35 | **0.48** | 0.91 |

**Table 6.** Latency and spatial and temporal sampling of the considered precipitation products.

| Product | Reference time step (min) | Latency (min) | Spatial resolution (km) |
|---|---|---|---|
| CML | 15 | 20 | 5 |
| Radar raw | 5 | 15 | 1 |
| Radar adj. | 60 | 60 | 1 |
| Rain gauges raw | 60 | 60 | - |
| ERG5 | 60 | 1440 | 5 |