# Peer review of "Commercial Microwave Links as a tool for operational rainfall monitoring in Northern Italy"

_Atmospheric Measurement Techniques, 2020_

## Referee Comment (RC1) · Anonymous Referee #1 · 27 Mar 2020

Review of the paper "Commercial Microwave Links as a tool for operational rainfall monitoring in Northern Italy" submitted to AMTD by Roversi et al. in 2020.

This manuscript is the first one to analyze the performance of commercial microwave links (CMLs) with a large data set (350 CMLs) in Italy. The applied processing algorithm is based on an open-source state-of-the-art method (RAINLINK) which has only been modified slightly. The main contribution of the manuscript hence is the analysis and discussion of the unique CML data set for the target region. The authors also give some new insights into how CML rainfall estimates perform under certain circumstances. Since CML data is not easily available for analysis in most countries the current manuscript provides an important contribution to the scientific community working on rainfall estimation and is certainly interesting for readers of AMT.

[Figure]

The manuscript is well structured and the applied processing methods are sound. Writing should be improved, though. I also have two major general comments regarding the analysis of the results which will require a major revision of the manuscript. I also have some minor and specific comments.

Major general comments and suggested major changes:

1. The main limitation of this study is that it lacks comparability to other studies because the quantitative analysis of the skill of the produced CML rainfall maps is only carried out for a subset of the data, namely the data pairs where the reference and the CML rainfall is > 0.1 mm/h. None of the other studies that use the RAINLINK algorithm and similar CML data (15 minute min/max) use this approach (see Table A1 in de Los et al. 2019 DOI: 10.1175/JTECH-D-18-0197.1). This also limits the interpretability of the results in this manuscript since the effect of bad FAR and POD, which lead to overestimation (high FAR) and underestimation (low POD), cannot be studied in the resulting rainfall fields. I strongly suggest to carry out the analysis of the rainfall fields for different subsetting variations. The most commonly used ones for comparing rainfall maps seem to be: 1. No threshold 2. Reference > 0.1 mm. This does not mean that all the plots have to be done several times, but at least the main skill metrics should be provided for the different subsets.

2. Since a large part of the quantitative analysis is based on interpolated rainfall maps, I strongly suggest to show several examples of interpolated CML rainfall maps, e.g. of one or two specific events and e.g. accumulated over the whole period.

Minor general comments:

- Choice of POD and FAR: I assume (since it is not specified in the manuscript I looked at other papers that use RAINLINK) that POD is hits/(hits+misses) and FAR is false_alarm/(hits+false_alarm). If this is the case, POD is the true positive rate (TPR). Wouldn't it than be better to use the false positive rate (FPR), like it is used in the ROC curve, instead of FAR. FPR and TPR are both normalized by the reference conditions.

FAR instead is normalized by the predicted positive conditions. Can you elaborate on this choice?

- The writing needs improvement throughout the manuscript, in particular the introduction and conclusion. Hence, I stopped very early to note down technical corrections and suggestions for stylistic improvements when reading the manuscript.

Specific comments:

L28: It would be good to have another or additional reference for the claim that the "last generation polarimetric systems have only partially mitigated" the radar QPE problems. The book by Ryzhkov and Zrnic, 2019 is certainly a very valuable textbook, but is is hard to find this conclusion in a 450 page reference. Access to it might also be limited.

L32: "...accuracy is still under evaluation (Tan et al., 2018)...". This statement is a bit weak. In addition, there are many studies that evaluate the performance of IMERG, also on a broader level than Tan et al., 2018.

L34: I think "broad diffusion" is the wrong term here. Something like "ubiquity" would fit better.

L36: "Accurate algorithms were introduced to measure ... drop size distribution ... water content". Since the sentence before talks about CMLs, the used references do not fit here, since they did not use, or only partly used, CML data. Dual-frequency and dual-polarization data, as used in the references, is mostly not (yet) available in operational CML networks.

L38: "...a spatially continuous rainfall path..." It is not clear to me what that means. Please rephrase.

L50: This is a very long and confusing sentence.

Section 2.1: What is the power quantization of P_min and P_max? Please specify.

Fig1. and section 2.1.1: Are there pixels without a CML, i.e. LC = 0. This is not clear

from the map and the text. Please clarify. If yes, what are the implications. E.g. if you would have to interpolate a rainfall field over two empty pixels in the west of Parma that would decrease the performance a lot compared to pixels that at least have one CML.

L170: It would be good to know what "set of consistency checks" has been used. Is everything done as in Overeem et al 2016? Even if yes, a short summary (2-3 sentences) would be good so that the reader does not have to go through the explanation in the reference.

L174: What is "a comparable decrease"? Please be more specific.

L178: Also here, it would be good to get more info on the outlier filters. What exactly was done? And even more important. How much data was removed?

L186: Did you use specific a and b values from van Leth et al, 2018? If not, it is not clear why this is cited here. Please cite the source of the a and b values.

L199: What was the length of the CMLs below 10 GHz? Even at 30 km (the maximal length in your data set according to section 2) a CML with 5 GHz is very insensitive to rainfall (approx. 0.05 dB at 1 mm/h path-averaged rainfall) so that light to moderate rain might not cause a detectable signal. Can you make sure that this does not have negative effects on the rainfall fields for light and moderate rainfall events (in the range 1-10 mm/h)? Couldn't it be that CMLs with zero rain rate are introduced in the interpolation method, which would better be left out? How much CMLs would you loose if you do not include CMLs below 10 GHz and how much does the "spatial coverage" decrease?

L209: My feeling as a non-native speaker is that "delineate" is the wrong term here.

Section 3.3: It is not clear from which reference you took which skill indicator. In my opinion, it would be best to define the skill indicators here to avoid any misconceptions.

L214: Complicated sentence and unclear formulation. I guess you are trying to say that your CML and reference products have a lot of rain rate that are so low that they can be neglected in any application.

Fig. 2: It would be good to know the minimal distance from the individual CMLs to the reference rain gauge.

Fig. 3: This figure contains a lot of useful information. It is a bit unstructured, though. It could be cleaned up by aligning the x-axis of each column and by sharing the legend in row 1 and 2. In row 3 two columns for "back" and "forward" could be used. Reusing the colors from row 2 in row 3 for different variables is also not ideal. The x-axis tick labels are also different in row 3 from row 1 and 2, so that it is not clear if the depicted periods are exactly the same. Hence, in particular the alignment of the x-axis would help. If you redo the plot, which is what I would suggest, than you could also reconsider the order of the rows. I feel that starting with the raw data (now row 2) would make more sense since this follows the CML data processing workflow. The meaning of the pink horizontal band, explained in the text, should also be explained in the figure caption.

L262: Can the overestimation of the one CML be explained by the spatial distance between this CML, the other CMLs and the gauge? If not, what is your explanation?

L269: Does this CML show these differences between P_max and P_min during the whole period? If yes, are there other CMLs that show something similar? Do you have any explanation or mitigation strategy? Regarding an operational application of CMLs there should be a way to deal with this kind of signals.

L281: If I understand correctly this data set is not part of the data set for the main analysis of the paper, correct? Please clarify in the text.

Section 4.1.3: It would be important to know the height of the antennas and the estimated height of the melting layer or zero-degree level. Form the fact that the data is from the month of March is cannot be concluded that the CML measured mixed-phase or solid precipitation.

L288: "...'bright band' in the radar reflectivity maps and is thus easily detected". If you have a dual-pol radar with a working hydrometeor classification, then yes, it can be

detected. If not, than this is quite hard to do for smaller scale precipitation events and on short temporal durations. I suggest to add some more details to the explanation in the text.

L296: Is this underestimation due to missed event or to a general underestimation of the CML rain rates? And why didn't you try to adjust the wet antennae compensation to compensate this underestimation? Overeem et al. 2013 and 2016 calibrated the wet-antenna compensation for a specific subset of their data, so it might neither be optimal nor applicable to your data. Please explain.

L298: Since your study and the two other studies all use a different "Filter" (see your Table 2) the results are not really comparable. In particular your choice of "Ref. AND Product > 0.1 mm h$-1$" neglects the negative effects of false positives and false negatives. See my major comment above.

L318: "The accuracy in the estimates is reached at the expense of POD, ETS and BIAS: around 50% of the rainfall duration is lost in this are". I understand that when FAR is lower (mentioned in the sentence before) and POD is lower there are less rain events, both correct and incorrect ones, in the resulting CML rainfall time series. That would explain that there is even more tendency to underestimate here. But, if I understood correctly, the bias is only calculated from values where both CML and reference are above 0.1 mm/h, so that false and missed CML rain events have no impact on the calculation of the bias. Can you elaborate on that?

L324: Remove the "For" at the beginning of the sentence

L327: "...this suggests that LC is probably not the only variable at play there". This is good to know, since that would have meant that regions with high CML density perform bad with the used algorithm. The CML data set of Overeem et al 2016 also has regions with a very dense network and regions with a coarser network. Hence, a strong dependence of the RAINLINK algorithm on LC should have already been noticed by them. Could it be that there is one CML in this area that shows "strange" behavior, e.g.

strong fluctuations, that negatively affects the POD of the many surrounding CMLs by not letting RAINLINK do the detection of rain events?

L355: Since your reference data set ERG5 is an interpolated rain gauge product, it might miss small scale rain events compared to the radar. Assuming that clutter removal was done in a sufficiently good way, the radar should not have a high FAR in general. Couldn't the fact that ERG5 might miss some real rain events explain the high FAR of the radar product?

L362: "...making CML a more robust sensor." Robust in what sense? Please explain in more detail in the text.

L365: When speaking about the "operational context" and the advantages of CMLs it should be discussed how the low POD, found in this study, affects the CML's potential for operational applications. This should be part of this paragraph.

---

## Referee Comment (RC2) · Anonymous Referee #2 · 1 Apr 2020

Review of the paper "Commercial Microwave Links as a tool for operational rainfall monitoring in Nothern Italy"

The manuscript aims to evaluate the potential of rainfall retrieval from CML network at regional scale in northern Italy to create rainfall maps for operational purposes. The paper describes the unique data set from the region of interest. It should be noted here that the collection of CML data set in large telecommunication network is still a challenge. However, from global perspective other studies already described similar experiments with identical (or wider) scales employing CML data using more advanced methods. Main weakness of the manuscript is therefore related to CML data processing and data analysis which is based on open-source package RAINLINK applied on CML data in northern Italy. I am missing the definition and answering the important re-

search questions which can provide new insights in CML rainfall retrieval. The overall scientific significance of the manuscript is fair.

Therefore, the manuscript needs major revisions. I see several aspects that can be studied using such data set. The quality of CML product is questionable and it show systematic underestimation. Then one way could be to test/develop other processing methods of CML data to reduce this bias and improve the quality of the product. Other interesting point could be an orographic aspect which is mentioned in the manuscript, but not studied in detail.

General comments:

1. The results show systematic underestimation of QPE derived from CMLs. RAINLINK package contains several strong assumptions (constant WAA of 2.3 dB, constant k-R parameters etc. ) which can influence the results significantly. Recent knowledge shows that WAA is complex process with many unknowns (e.g. Leth et al., 2018). The dataset probably contains a certain portion of sensors with low sensitivity (this is reviewer assumption since the CML statistic is not provided) to rainfall where WAA can play dominant role in resulting rainfall retrieval. I would recommend to make at least sensitivity analysis of the results to most significant parameters.

2. Spatial interpolation is based on assumption the path-integrated rainfall is represented as a point measurement. This assumption can be used for rough grid 5x5 km and shorter CMLs. However, it is weak for single link comparison (section 4.1) including single event comparison. Here, spatial-temporal structure of rain together with the layout of given RG and CMLs can play significant role. Then it is impossible to compare single point measurements and CMLs observations.

3. Since rainfall maps are the key product of the presented study, I would expect to show visually CML rainfall maps – event-based or cumulative rainfall compared to reference.

[Figure]

4. I am missing relevant discussion section in the paper

5. I am not satisfied with the conclusions which do not provide novel information beyond the state of the art in the field of CML rainfall retrieval.

Specific comments:

L. 33-45. I don't agree with this paragraph since the first sentence refer to CMLs. The provide references are partly based on experimental microwave link setup, not CMLs. I wonder we know accurate algorithms for DSD, water content etc. based on CML observation.

L. 70. observation period – since later in the manuscript some analysis are event based I would add into the Supplementary material information and data about precipitation events during observation period. For selected rainfalls and locations used later in section 4.1 some detailed rainfall metrics would be welcome.

L. 90-94 The usage of CMLs with low operating frequencies 6 – 15 GHz is questionable for QPE because of low sensitivity of those devices to rainfall even with longer path lengths. It would be useful to provide statistic evidence of different frequency bands in the data set including calculated theoretical sensitivity to rainfall. Then the effect of constant WAA to the results would be much clearer.

L. 104 Spatial distribution of LC – could you explain why the LC is lower in the main regional cities (Parma and Bologna) than in countryside – Figure 1?

Section 2.1.2 Transmitting power levels I found this paragraph a little bit confusing. I would ask to rephrase it to provide clear information about ATCP processing

L. 193 Interpolation – please explain how path-averaged rainfall depth from each CML is implemented into spatial interpolation. This not very clear from provided description

Section 4.1 Single link verification – see my general comment about point and path-averaged rainfall estimates. This is difficult to understand especially when we don't

see detailed information about precipitation metrics during observation period. The data also does not correspond with previous statement in Section 2, that in higher altitudes are higher amount of rains.

L. 228 I suggest this statement as weak and confusing "They have been chosen in areas with different terrain and network density and far from the cities, as CMLs in urban areas are already well studied and also the most eligible to be replaced by optic fibres." I don't see why CMLs in cities should work in different way than in country side. Is there evidence that CML in cities are already well studied and in the countryside not? Network development is not relevant for this paper and this sentence is speculation.

Sections 4.1.1.-4.1.2. - Best and Worst Case Example – I do not understand why there is no text information and results interpretation with respect to rainfall intensity and rainfall characteristics. 4.1.2 represents light rain when the sensitivity to rainfall of CMLs is low. WAA is significant here anyway. Also, data provided from NMS system in form Pmin Pmax are limiting factor. This shows clear limits of CML for light rainfalls and Pmin Pmax approach.

Section 4.1.3 – I do not think that this melting layer story fits to this story. First, the data set is presented as spring – summer period. The article is focused on liquid precipitation, this is another story.

L. 320-330 I do not fully agree with those statements about LC. Different LC often means different frequency bands distribution. In the region with high LC one can expect higher frequency bands with higher rainfall sensitivity.

Figures general – I found inconsistency when using brackets for units – none, () or [] in different figures

Figure 7. I do not understand the "bad" results of adjusted radar in comparison to the reference which was used for radar adjustment. The results are comparable to unadjusted radar data. Could you explain that?

---

## Author Comment (AC1) · 20 May 2020

**Review of the paper "Commercial Microwave Links as a tool for operational rainfall monitoring in Northern Italy" submitted to AMTD by Roversi et al. in 2020.**

**This manuscript is the first one to analyze the performance of commercial microwave links (CMLs) with a large data set (350 CMLs) in Italy. The applied processing algorithm is based on an open-source state-of-the-art method (RAINLINK) which has only been modified slightly. The main contribution of the manuscript hence is the analysis and discussion of the unique CML data set for the target region. The authors also give some new insights into how CML**

[Figure]

**rainfall estimates perform under certain circumstances. Since CML data is not easily available for analysis in most countries the current manuscript provides an important contribution to the scientific community working on rainfall estimation and is certainly interesting for readers of AMT.**

**The manuscript is well structured and the applied processing methods are sound. Writing should be improved, though. I also have two major general comments regarding the analysis of the results which will require a major revision of the manuscript. I also have some minor and specific comments.**

*We thank the reviewer for the positive introductory remarks and the careful review of the manuscript. We addressed all her/his comments point-by-point in italic, outlining how we are modifying the manuscript.*

**Major general comments and suggested major changes:**

**1. The main limitation of this study is that it lacks comparability to other studies because the quantitative analysis of the skill of the produced CML rainfall maps is only carried out for a subset of the data, namely the data pairs where the reference and the CML rainfall is > 0.1 mm/h. None of the other studies that use the RAINLINK algorithm and similar CML data (15 minute min/max) use this approach (see Table A1 in de Los et al. 2019 DOI: 10.1175/JTECH-D-18-0197.1). This also limits the interpretability of the results in this manuscript since the effect of bad FAR and POD, which lead to overestimation (high FAR) and underestimation (low POD), cannot be studied in the resulting rainfall fields. I strongly suggest to carry out the analysis of the rainfall fields for different subsetting variations. The most commonly used ones for comparing rainfall maps seem to be: 1. No threshold 2. Reference > 0.1 mm. This does not mean that all the plots have to be done several times, but at least the main skill metrics should be provided for the different subsets.**

*We agree that for the comparison with the previous studies a set of indicators with the*

*same (or similar) settings is needed. We will compute continuous indicators for the set with no threshold and with Reference > 0.1 mm/h and we will add them to Table 2 in the new manuscript. For the categorical scores instead no filtering was performed. The threshold of 0.1 mm/h was used there to discriminate wet and dry samples in the confusion matrix. We acknowledge that it could have been unclear from the text and we will rephrase the paragraph for better understandability.*

**2. Since a large part of the quantitative analysis is based on interpolated rainfall maps, I strongly suggest to show several examples of interpolated CML rainfall maps, e.g. of one or two specific events and e.g. accumulated over the whole period.**

*We welcome this suggestion and we will add a section where two case studies are reported (e.g. the best and the worst), including the discussion of meteorological conditions, reporting interpolated maps and indicators.*

**Minor general comments:**

**- Choice of POD and FAR: I assume (since it is not specified in the manuscript I looked at other papers that use RAINLINK) that POD is hits/(hits+misses) and FAR is falsealarm/(hits+falsealarm). If this is the case, POD is the true positive rate (TPR). Wouldn't it than be better to use the false positive rate (FPR), like it is used in the ROC curve, instead of FAR. FPR and TPR are both normalized by the reference conditions. FAR instead is normalized by the predicted positive conditions. Can you elaborate on this choice?**

*The reviewer understood correctly: POD=hit/(hits+misses) and FAR=falsealarm/(hits+falsealarm). This choice to favour FAR over FPR derives from its common use in deterministic precipitation forecast/estimate validation (Tang et al., 2020; Petracca et al., 2018; Puca et al., 2013, McBride and Ebert, 1998, among others). The FPR= falsealarm/(falsealarm+correctnegatives), more common in probabilistic forecast verification, is heavily influenced by the most populated category*

*(correctnegatives): in case of small scale (or rare) rain pattern, FPR can decrease without any skill in the forecast since the no-rain condition is the most common in the target area. FPR, for the same reason, could also be misleading when different seasons/climates, with different rain occurrence, are compared.*

**- The writing needs improvement throughout the manuscript, in particular the introduction and conclusion. Hence, I stopped very early to note down technical corrections and suggestions for stylistic improvements when reading the manuscript.**

*We will improve the revised manuscript through a careful review of the language.*

**Specific comments:**

**L28: It would be good to have another or additional reference for the claim that the "last generation polarimetric systems have only partially mitigated" the radar QPE problems. The book by Ryzhkov and Zrnic, 2019 is certainly a very valuable textbook, but is is hard to find this conclusion in a 450 page reference. Access to it might also be limited.**

*We agree with the Reviewer and we'll include few references more focused on the QPE of polarimetric radar performance evaluation (Figueras i Ventura et al., 2012; Gou et al., 2019; Cocks et al., 2019).*

*Figueras i Ventura J, Boumahmoud A-A, Fradon B, Dupuy P, Tabary P. 2012. Long-term monitoring of French polarimetric radar data quality and evaluation of several polarimetric quantitative precipitation estimators in ideal conditions for operational implementation at C-band. Q. J. R. Meteorol. Soc. 138: 2212–2228. DOI:10.1002/qj.1934*

*Gou, Y.; Chen, H.; Zheng, J. Polarimetric Radar Signatures and Performance of Various Radar Rainfall Estimators during an Extreme Precipitation Event over the Thousand-Island Lake Area in Eastern China. Remote Sens. 2019, 11, 2335. https://doi.org/10.3390/rs11202335*

*Cocks, S., L. Tang, P. Zhang, A. Ryzhkov, B. Kaney, K. L. Elmore, Y. Wang, J. Zhang, and K. Howard, 2019: A prototype quantitative precipitation estimation algorithm for operational S-band polarimetric radar utilizing specific attenuation and specific differential phase. Part II: Performance verification and case study analysis. J. Hydrometeor., 20, 999–1014, https://doi.org/10.1175/JHM-D-18-0070.1.*

**L32: ": : :accuracy is still under evaluation (Tan et al., 2018): : :". This statement is a bit weak. In addition, there are many studies that evaluate the performance of IMERG, also on a broader level than Tan et al., 2018.**

*Of course, many papers are dealing with satellite product validation, but very few of them deal with high resolution (hourly) data, mostly focusing on daily to annual integrals. We rewrite more precisely the sentence "their accuracy is difficult to assess at high spatial and temporal scales, depending on local climatology (Tang et al., 2020)", and included a more recent and pertinent reference.*

*Tang, G., M. P. Clark, S. M. Papalexiou, Z. Ma, Y. Hong, Have satellite precipitation products improved over last two decades? A comprehensive comparison of GPM IMERG with nine satellite and reanalysis datasets, Remote Sensing of Environment, Volume 240, 2020, 111697, https://doi.org/10.1016/j.rse.2020.111697.*

**L34: I think "broad diffusion" is the wrong term here. Something like "ubiquity" would fit better.**

*"ubiquity" means everywhere, including ocean and desert, we'll cut the adjective, leaving: ". . .in the last decades with the diffusion of microwave. . .".*

**L36: "Accurate algorithms were introduced to measure : : : drop size distribution : : : water content". Since the sentence before talks about CMLs, the used references do not fit here, since they did not use, or only partly used, CML data. Dual-frequency and dual-polarization data, as used in the references, is mostly not (yet) available in operational CML networks.**

*The reviewer is right: the sentence was not correctly contextualized, and we reworded to: "Accurate experiments and numerical simulation were used to assess the capability of microwave links to measure average rainfall rates (Rahimi et al., 2003), drop size distribution (Rincon and Lang, 2002; van Leth et al., 2020) and water content (Jameson, 1993). On the same token, the possibility to have a spatially continuous rainfall pattern depends on the density and distribution of the links, making this approach of particular interest for urban areas. . . ".*

**L38: ": : :a spatially continuous rainfall path: : :" It is not clear to me what that means. Please rephrase.**

*The word "path" is replaced by "pattern".*

**L50: This is a very long and confusing sentence.**

*We'll remove this sentence in the new version of the manuscript.*

**Section 2.1: What is the power quantization of Pmin and Pmax? Please specify.**

*The quantization for Power is 1dB, we'll report this number in the revised manuscript.*

**Fig1. and section 2.1.1: Are there pixels without a CML, i.e. LC = 0. This is not clear from the map and the text. Please clarify. If yes, what are the implications. E.g. if you would have to interpolate a rainfall field over two empty pixels in the west of Parma that would decrease the performance a lot compared to pixels that at least have one CML.**

*We changed the colour scale of Figure 1, to make clear the presence of few LC=0 grid boxes. Nevertheless, we do not think that cells with LC=0 represent an issue because we aim to evaluate an interpolated product whose goal is precisely filling the empty gaps between separate measurements. Previous CML papers also show rainfall maps interpolated at a finer scale (1 km) and with sparser and more inhomogeneous CML networks (e.g. Overeem et al., 2016). Besides, we agree that better results are likely to be expected from regions with higher coverage. We already address the matter*

*throughout the analysis of the LC dependency.*

**L170: It would be good to know what "set of consistency checks" has been used. Is everything done as in Overeem et al 2016? Even if yes, a short summary (2-3 sentences) would be good so that the reader does not have to go through the explanation in the reference.**

*The consistency criteria require that: the frequency is inside a specified range; there are no multiple occurrences for the same ID and DateTime, every ID has always the same geographical coordinates, not-available (NA) entries are not present. We will add a sentence that clarifies this point and possibly add some other algorithm information in the Supplement.*

**L174: What is "a comparable decrease"? Please be more specific.**

*Wet-Dry Classification is described in Appendix C of Overeem et al. 2016 and we used exactly their procedure. The description was here treated only qualitatively on purpose: we will add an explicit reference to the appendix and we will specify that the values inside of the NLA classification are left unchanged.*

**L178: Also here, it would be good to get more info on the outlier filters. What exactly was done? And even more important. How much data was removed?**

*We will add some additional details on the procedure. Also, statistics on outliers will be added to the revised version.*

**L186: Did you use specific a and b values from van Leth et al, 2018? If not, it is not clear why this is cited here. Please cite the source of the a and b values.**

*Van Leth et al.(2018) is cited here only to support the assertion about which variables the a and b parameters are sensitive to. A proper description of which parameters are utilized in our work is reported in the following Section 3.2. We will add some internal references to make that clearer.*

**L199: What was the length of the CMLs below 10 GHz? Even at 30 km (the maximal length in your data set according to section 2) a CML with 5 GHz is very insensitive to rainfall (approx. 0.05 dB at 1 mm/h path-averaged rainfall) so that light to moderate rain might not cause a detectable signal. Can you make sure that this does not have negative effects on the rainfall fields for light and moderate rainfall events (in the range 1-10 mm/h)? Couldn't it be that CMLs with zero rain rate are introduced in the interpolation method, which would better be left out? How much CMLs would you loose if you do not include CMLs below 10 GHz and how much does the "spatial coverage" decrease?**

*This is a very important point and we thank the reviewer for having it highlighted. In our network, we have only five links between 5 and 10 GHz. We will remove them and we do not expect any major change in the results. We also are investigating the sensitivity for all links, and we'll discuss this in the revised paper.*

**L209: My feeling as a non-native speaker is that "delineate" is the wrong term here.**

*We replaced "delineate" with "detect".*

**Section 3.3: It is not clear from which reference you took which skill indicator. In my opinion, it would be best to define the skill indicators here to avoid any misconceptions.**

*We'll add the description of the indicators to be clearer.*

**L214: Complicated sentence and unclear formulation. I guess you are trying to say that your CML and reference products have a lot of rain rate that are so low that they can be neglected in any application.**

*The reviewer understood correctly. However, we reworded the sentence to "Both interpolated CML and reference field have a large number of very low positive values (below 0.1 mm h-1) that are not of interest in any application, but which are potentially*

*very influential in normalized error metrics".*

**Fig. 2: It would be good to know the minimal distance from the individual CMLs to the reference rain gauge.**

*We added, as further information on the figure, the minimum distance between gauge and link. Discussions around this distance are also going to be added to Section 4.1 and following.*

**Fig. 3: This figure contains a lot of useful information. It is a bit unstructured, though. It could be cleaned up by aligning the x-axis of each column and by sharing the legend in row 1 and 2. In row 3 two columns for "back" and "forward" could be used. Reusing the colors from row 2 in row 3 for different variables is also not ideal. The x-axis tick labels are also different in row 3 from row 1 and 2, so that it is not clear if the depicted periods are exactly the same. Hence, in particular the alignment of the x-axis would help. If you redo the plot, which is what I would suggest, than you could also reconsider the order of the rows. I feel that starting with the raw data (now row 2) would make more sense since this follows the CML data processing workflow. The meaning of the pink horizontal band, explained in the text, should also be nna explained in the figure caption.**

*We are considering the re-design of Figure 3, following the reviewer's advice. We were aware that this figure could appear unclear, and thank the reviewer for the suggestions.*

**L262: Can the overestimation of the one CML be explained by the spatial distance between this CML, the other CMLs and the gauge? If not, what is your explanation?**

*We'll include the distance information in the revised manuscript, and consider also other causes of uncertainties in the matching.*

**L269: Does this CML show these differences between Pmax and Pmin during the whole period? If yes, are there other CMLs that show something similar?**

**Do you have any explanation or mitigation strategy? Regarding an operational application of CMLs there should be a way to deal with this kind of signals.**

*This is to our understanding a common feature to all CMLs and, more generally, to the 15 minutes MinMax sampling strategy. We will say that more clearly in the revised manuscript. We will also address the connection of this feature to the manual ATPC correction which involves only Pmin.*

**L281: If I understand correctly this data set is not part of the data set for the main analysis of the paper, correct? Please clarify in the text.**

**Section 4.1.3: It would be important to know the height of the antennas and the estimated height of the melting layer or zero-degree level. Form the fact that the data is from the month of March is cannot be concluded that the CML measured mixed-phase or solid precipitation.**

**L288: ": : :"bright band in the radar reflectivity maps and is thus easily detected". If you have a dual-pol radar with a working hydrometeor classification, then yes, it can be detected. If not, than this is quite hard to do for smaller scale precipitation events and on short temporal durations. I suggest to add some more details to the explanation in the text.**

*Of course, we have polarimetric data observations and assessed the presence of bright band without any doubt. However, after the suggestion of the other reviewer also, we decided to drop this section, being a little out of the main objective of the work.*

**L296: Is this underestimation due to missed event or to a general underestimation of the CML rain rates? And why didn't you try to adjust the wet antennae compensation to compensate this underestimation? Overeem et al. 2013 and 2016 calibrated the wet-antenna compensation for a specific subset of their data, so it might neither be optimal nor applicable to your data. Please explain.**

*The first question could be addressed by comparing the overall values ME (-26%)*

*indicating the relative deficit of measured rain amount with the MB (0.7), the relative occurrence of estimated wet samples with respect to the real number of wet samples. The underestimate seems to affect the 30% the number of "events" and a little bit less (26%) the amount of water. From the conceptual point of view, however, the two things are tightly connected: the underestimation of the rainrate results in an underestimation of rain occurrence, as soon as the underestimate affects rainrate values just above the threshold. We will add a comment on this.*

*As for the second issue, our feeling is that the reference data we considered (used in operational offices) are not suitable to be used as calibrator, in term of quality and spatial and temporal characteristics, as also the other reviewer remarked. Anyway, we are running some trials with decreasing Aa, and will report and discuss their results in the revised manuscript. We already show how an overestimation of Aa could affect the algorithm (figure 3), and indicate the Aa high value as a possible issue to address, once an experimental facility with the necessary accuracy would become available.*

**L298: Since your study and the two other studies all use a different "Filter" (see your Table 2) the results are not really comparable. In particular your choice of "Ref. AND Product > 0.1 mm h-1" neglects the negative effects of false positives and false negatives. See my major comment above.**

*We agree with the reviewer and we will provide a new set of indicators' values suitable for comparisons with previous works, and added a column in Table 2.*

**L318: "The accuracy in the estimates is reached at the expense of POD, ETS and BIAS: around 50% of the rainfall duration is lost in this area". I understand that when FAR is lower (mentioned in the sentence before) and POD is lower there are less rain events, both correct and incorrect ones, in the resulting CML rainfall time series. That would explain that there is even more tendency to underestimate here. But, if I understood correctly, the bias is only calculated from values where both CML and reference are above 0.1 mm/h, so that false and missed CML**

**rain events have no impact on the calculation of the bias. Can you elaborate on that?**

*First, we made a mistake: "BIAS" (undefined in this work) stands for Multiplicative Bias (MB), i.e. numberofestimatedwet)/(numberofobservedwet). Hence a Multiplicative Bias of 0.47 indicates that only half of the wet samples are found, where for wet we mean rain depth > 0.1 mm/h (both estimated and observed). However, the amount of rain lost in this area (given by ME) is similar to other areas, and the indicators of numerical accuracy of the estimates (CV and CC), computed on wet-wet samples, are quite high. This indicates that in this area the rainrate is estimated with higher accuracy, while the discrimination wet/dry is worse. Categorical scores are calculated on the unfiltered datasets, around the filtering threshold.*

**L324: Remove the "For" at the beginning of the sentence**

*Ok, thanks.*

**L327: ": : :this suggests that LC is probably not the only variable at play there". This is good to know, since that would have meant that regions with high CML density perform bad with the used algorithm. The CML data set of Overeem et al 2016 also has regions with a very dense network and regions with a coarser network. Hence, a strong dependence of the RAINLINK algorithm on LC should have already been noticed by them. Could it be that there is one CML in this area that shows "strange" behavior, e.g.strong fluctuations, that negatively affects the POD of the many surrounding CMLs by not letting RAINLINK do the detection of rain events?**

*Overeem et al. (2016) showed how the CML performance varies against the mean link density (our "LC") by analysing normalized variance and correlation on a 74km2 grid. Our results for CV and CC estimated at 25 km2 are in good agreement with them. In addition to their work, we show also the effect of LC on the categorical indicators, providing some interesting results for the FAR especially and giving more insight into the*

*topic in general. However, we were not able to isolate all the sources of uncertainties and to gauge the performances of the single links individually. We'll better express this peculiarity of our work in the revised manuscript.*

**L355: Since your reference data set ERG5 is an interpolated rain gauge product, it might miss small scale rain events compared to the radar. Assuming that clutter removal was done in a sufficiently good way, the radar should not have a high FAR in general. Couldn't the fact that ERG5 might miss some real rain events explain the high FAR of the radar product?**

*The clutter is removed through a static map of clutter, a beam trajectory simulation and an anomalous propagation cancellation, (see Fornasiero et al., 2006). Moreover, WiFi/WiMax signals are filtered through a decision tree and a fuzzy logic techniques which expoitZ, Zdr, W, V and Z and Zdr variance. We do not think therefore that the clutter is the reason for the high FAR. We suppose instead, as the reviewer pointed out, that a reason for high false alarms ratio could be that ERG5 misses some small scale events. We'll modify the sentence: "...while rain gauge (as well as the reference product ERG5) and CML networks...".*

**L362: ": : :making CML a more robust sensor." Robust in what sense? Please explain in more detail in the text.**

*We want to point out that to fully exploit radar capabilities a proper Z-R relationship should be used, while the CML k-R relation is almost not dependent on DSD due to its linearity in this frequency range (Leijnse et al., 2008). We added few words to the sentence: "thus making CML a more robust sensor, in the sense that the coefficients to retrieve rainrate do not depend on the DSD".*

bf L365: When speaking about the "operational context" and the advantages of CMLs it should be discussed how the low POD, found in this study, affects the CML's potential for operational applications. This should be part of this paragraph.

*We'll specify better the possible role of CML in the operational context, modifying the sentence to: "In an operational context, where several precipitation products (each one with its proper error structure) are available to the forecaster, it is of great relevance the latency of the precipitation product, i.e. the time taken from the acquisition of the basic data (the occurrence of the event) and the delivery of the product in a ready-to-use form." And at the end of the paragraph, we added the comment: "It is to the operators' preference, based on product accuracy and current meteorological conditions, to make use of the most suitable product".*

---

## Author Comment (AC2) · 20 May 2020

**The manuscript aims to evaluate the potential of rainfall retrieval from CML network at regional scale in northern Italy to create rainfall maps for operational purposes. The paper describes the unique data set from the region of interest. It should be noted here that the collection of CML data set in large telecommunication network is still a challenge. However, from global perspective other studies already described similar experiments with identical (or wider) scales employing CML data using more advanced methods.**

*We thank the reviewer for the comments and for the indications to improve our work. We'll modify the manuscript as outlined below, replying point by point in italic.*

[Figure]

**Main weakness of the manuscript is therefore related to CML data processing and data analysis which is based on open-source package RAINLINK applied on CML data in northern Italy. I am missing the definition and answering the important research questions which can provide new insights in CML rainfall retrieval. The overall scientific significance of the manuscript is fair. Therefore, the manuscript needs major revisions. I see several aspects that can be studied using such data set. The quality of CML product is questionable and it show systematic underestimation. Then one way could be to test/develop other processing methods of CML data to reduce this bias and improve the quality of the product. Other interesting point could be an orographic aspect which is mentioned in the manuscript, but not studied in detail.**

*We agree with the reviewer that in this paper we do not address basic research questions, such as to set-up advanced algorithms or tackle challenging issues, but we think that one important task in the research activity is to communicate to potential users possible applications of the research itself.*

*Moreover, in our opinion, the data available to us was simply not accurate and complete enough to develop and test new algorithms or to analyse the impact of orography or other known critical aspects of the rain retrieval from CMLs. Longer data time series over wider regions and a more reliable and representative reference dataset is needed to do such studies, which was not accessible to us at the time. However, we believe that this work demonstrates a good potential of the technology even at its most basic implementation, and gives valuable hints for future regional improvements.*

*The objective of our study, indeed, was to test the possible role of CML retrievals in an operational environment without any previous study on the characteristics of the available CMLs. We performed an "out of the box" approach as we aimed to test the performances obtainable without specific calibration (whose related effort could be not sustainable in many places). We assessed that a robust and freely available algorithm (such as RAINLINK) provides a product with spatial and temporal characteristics*

*comparable to products routinely available to the operators in our region. Moreover, we highlighted how the performances of RAINLINK can be improved, addressing the few parameters that could benefit from a calibration/validation campaign (with proper instruments), once it will become possible.*

*To clarify, we'll modify the sentences in the revised manuscript at lines 58-64 as follows: "... CML attenuation data, using a well-established, freely-available algorithm (i.e. RAINLINK, Overeem et al. (2016a)), over two areas of interest in the Po Valley (provinces of Bologna and Parma), where CML data have been obtained from Vodafone (direct purchase). Both areas contain river basins of considerable local interest, which will be addressed specifically. The further aim of the validation study is to set the background for possible inclusion of CML data in the operational routine procedures for precipitation monitoring in the Meteorological Service of the Regional Agency for Environmental Protection and Energy (Arpae-SIMC), showing baseline potential of the methodology and indicating a direction toward which to direct the implementation and tuning effort."*

**General comments:**

**1. The results show systematic underestimation of QPE derived from CMLs. RAINLINK package contains several strong assumptions (constant WAA of 2.3 dB, constant k-R parameters etc. ) which can influence the results significantly. Recent knowledge shows that WAA is complex process with many unknowns (e.g. Leth et al., 2018). The dataset probably contains a certain portion of sensors with low sensitivity (this is reviewer assumption since the CML statistic is not provided) to rainfall where WAA can play dominant role in resulting rainfall retrieval. I would recommend to make at least sensitivity analysis of the results to most significant parameters.**

*It is well known (van Leth et al., 2018 among many others) that antenna wetting is one of the main problems in microwave estimation of precipitation, and for this reason we*

*think we cannot address this issue with our operationally oriented verification system. We remark that to address this issue properly, van Leth et al. (2018) deployed a unique experimental setting, with extremely controlled antenna conditions (time-lapse camera pointing the antennas) and accurate reference measurements (five disdrometers along link path). Even with this unprecedented experimental setting, neither van Leth et al. (2018) could definitively address this issue in a general way.*

*To give some more hints to the reader interested in the use of RAINLINK, we'll add some trials we made, changing the fixed WAA threshold, and discuss the results.*

**2. Spatial interpolation is based on assumption the path-integrated rainfall is represented as a point measurement. This assumption can be used for rough grid 5x5 km and shorter CMLs. However, it is weak for single link comparison (section 4.1) including single event comparison. Here, spatial-temporal structure of rain together with the layout of given RG and CMLs can play significant role. Then it is impossible to compare single point measurements and CMLs observations.**

*We thank the reviewer for pointing out this issue and agree that the comparison between single link and single rain gauge is affected by many uncertainties. However, similar shortcomings should also apply when comparisons with other instruments are considered. As an example, radar data, which could be in principle preferred because of its spatially integrated nature, suffer by many other uncertainties that make the precise comparison with line integrated CML estimate questionable. The only way to proceed properly seems to be to follow the van Leth et al. (2020) approach or similar. Anyway, we believe that our analysis, could still give the reader some valuable hints to understand the behaviour of interpolated RAINLINK products used in the following sections of the paper. In the discussion, we'll consider the possible shortcoming of our analysis. We'll also add some statistics regarding CML-gauge distances which prove that the averaging of the rainrate along the path and the comparison with a nearby rain-gauge are assumptions of the same order of magnitude for what concerns the scale of*

*the variance of the precipitation field.*

**3. Since rainfall maps are the key product of the presented study, I would expect to show visually CML rainfall maps – event-based or cumulative rainfall compared to reference.**

*According to also the other reviewer, we'll show 2 case studies in the revised manuscript, with maps and skill indicators, and the cumulated map for the whole period.*

**4. I am missing relevant discussion section in the paper**

*We would prefer not to change the structure of the paper, but we will evaluate whether to deeper discuss the issues suggested by the reviewer point by point in the Results section or to add a separate Discussion section*

**5. I am not satisfied with the conclusions which do not provide novel information beyond the state of the art in the field of CML rainfall retrieval.**

*We'll better illustrate the results of our work, keeping in mind that we want to be at the application side of the problem.*

**Specific comments:**

**L. 33-45. I don0t agree with this paragraph since the first sentence refer to CMLs. The provide references are partly based on experimental microwave link setup, not CMLs. I wonder we know accurate algorithms for DSD, water content etc. based on CML observation.**

*The reviewer is right: the sentence was badly structured. We will rewrite as: "Accurate experiments and numerical simulation were used to assess the capability of microwave links to measure average rainfall rates (Rahimi et al., 2003), drop size distribution (Rincon and Lang, 2002; van Leth et al., 2020) and water content (Jameson, 1993). On the same token, the possibility to have a spatially continuous rainfall pattern depends on the density and distribution of the links, making this approach of particular interest*

*for urban areas. . . "*

**L. 70. observation period – since later in the manuscript some analysis are event based I would add into the Supplementary material information and data about precipitation events during observation period. For selected rainfalls and locations used later in section 4.1 some detailed rainfall metrics would be welcome.**

*We'll add to the Supplementary material the PDFs of rainrates to characterize precipitation during the observation period.*

**L. 90-94 The usage of CMLs with low operating frequencies 6 – 15 GHz is questionable for QPE because of low sensitivity of those devices to rainfall even with longer path lengths. It would be useful to provide statistic evidence of different frequency bands in the data set including calculated theoretical sensitivity to rainfall. Then the effect of constant WAA to the results would be much clearer.**

*We thank the reviewer for having pointed this topic out. The sensitivity characteristics of the CML network will be reported in the Supplementary material, and briefly summarized and discussed in the revised text. Low sensitivity links will be removed from the data set before updating the results, which are yet expected to remain almost unchanged.*

**L. 104 Spatial distribution of LC – could you explain why the LC is lower in the main regional cities (Parma and Bologna) than in countryside – Figure 1?**

*In Italy and generally in the world, most of the CMLs in urban areas are being substituted by underground optical fibre cables. We'll add a comment on this.*

**Section 2.1.2 Transmitting power levels I found this paragraph a little bit confusing. I would ask to rephrase it to provide clear information about ATCP processing.**

*We'll rephrase the paragraph, improving clarity and describing the manual correction of the ATPC in detail.*

**L. 193 Interpolation – please explain how path-averaged rainfall depth from each CML is implemented into spatial interpolation. This not very clear from provided description Section 4.1 Single link verification – see my general comment about point and pathaveraged rainfall estimates. This is difficult to understand especially when we don't see detailed information about precipitation metrics during observation period. The data also does not correspond with previous statement in Section 2, that in higher altitudes are higher amount of rains.**

*Interpolation of the point path-averaged rainfall estimates (placed in the middle of the links' paths) is performed through ordinary kriging with range, sill and nugget derived from seasonal spatial correlation analyses of two years of gauge data of our region. More details on the precipitation characteristics involved in the analyses of Section 4.1 will be added in the revised manuscript.*

*The sentence in Section 2 was related to the rainfall climatology of the region. The case study likely represents an exception to the climatology.*

**L. 228 I suggest this statement as weak and confusing "They have been chosen in areas with different terrain and network density and far from the cities, as CMLs in urban areas are already well studied and also the most eligible to be replaced by optic fibres." I don0t see why CMLs in cities should work in different way than in country side. Is there evidence that CML in cities are already well studied and in the countryside not? Network development is not relevant for this paper and this sentence is speculation.**

*CML network characteristics are different going from cities to the countryside. Specifically, in the cities there are fewer CMLs since most of them have been already replaced with optical fibre (see also the answer to the comment on L104 and the recent The Netherlands' situation reported for example in Overeem et al. (2016), Introduction section, 4th paragraph). We provide many references for metropolitan CML studies with short links. Moreover, implications of network's developments on operational retrieval*

*capabilities are among the most relevant topics in the CML field.*

**Sections 4.1.1.-4.1.2. - Best and Worst Case Example – I do not understand why there is no text information and results interpretation with respect to rainfall intensity and rainfall characteristics. 4.1.2 represents light rain when the sensitivity to rainfall of CMLs is low. WAA is significant here anyway. Also, data provided from NMS system in form Pmin Pmax are limiting factor. This shows clear limits of CML for light rainfalls and Pmin Pmax approach.**

*We thank the reviewer for this comment: we'll include a more detailed discussion on these results, addressing especially the type of precipitation and the Pmax-Pmin approach, and also considering previous reviewer's comments.*

**Section 4.1.3 – I do not think that this melting layer story fits to this story. First, the data set is presented as spring – summer period. The article is focused on liquid precipitation, this is another story.**

*The melting layer episode does not belong to the 2-month dataset used for the main study, but it was a standalone dataset obtained from Vodafone for preliminary checking. This event occurred in February when freezing level could reach the ground, especially on the hills. Since liquid precipitation at midlatitude originates from frozen hydrometeors, the bright band is a rather common feature in our regions and introduces errors in the radar estimates often difficult to correct. Anyway, we understand that this issue is a bit far from the mainline of the work, so, we decided to remove this subsection.*

**L. 320-330 I do not fully agree with those statements about LC. Different LC often means different frequency bands distribution. In the region with high LC one can expect higher frequency bands with higher rainfall sensitivity.**

*We did not find any correlation between frequency and LC to date, but we thank the reviewer for the hint. As anticipated, we will carry out a deeper analysis of the links' sensitivity and we will discuss it in the revised manuscript, also deepening the under-*

*standing of LC contribution to our sensing performances.*

**Figures general – I found inconsistency when using brackets for units – none, ()
or [] in different figures**

*Thank you for noticing this, we will fix this inconsistency.*

**Figure 7. I do not understand the "bad" results of adjusted radar in comparison
to the reference which was used for radar adjustment. The results are compara-
ble to unadjusted radar data. Could you explain that?**

*The adjustment is performed with gauges and not with the interpolated reference, as
specified in Section 2.2.3. The procedure matches the rainrates estimated over the
gauge locations but does not ensure the consistency of the whole radar field with the
gauge interpolated one, mostly because of the high spatial variance of the radar field
(as already discussed in Section 4.2.2, L356). Therefore discrepancies in the areal
averages are not only to be tolerated but also expected. Moreover, the spatial autocor-
relation of the G/R adjustment factor is even lower during convective events, leading
to a less effective correction. We'll mention this in the revised manuscript and we will
add some documentation regarding the radar adjustment statistics and performances
in the Supplementary Material.*

---

## Author Response (AR1)

**Author response and changes in the revised manuscript of "Commercial Microwave Links as a tool for operational rainfall monitoring in Northern Italy" by Giacomo Roversi et al. (2020)**

*We would like to thank both Anonymous Referees for the careful review, the comments, and the suggestions on how to improve our work. We report the responses to all their comments point-by-point in italic, also indicating how we modified the manuscript. The integral revised manuscript marked-up with track changes is attached below the answers.*

**Response to Anonymous Referee #1**

**Major general comments and suggested major changes:**

**1. The main limitation of this study is that it lacks comparability to other studies because the quantitative analysis of the skill of the produced CML rainfall maps is only carried out for a subset of the data, namely the data pairs where the reference and the CML rainfall is > 0.1 mm/h. None of the other studies that use the RAINLINK algorithm and similar CML data (15 minute min/max) use this approach (see Table A1 in de Los et al. 2019 DOI: 10.1175/JTECH-D-18-0197.1). This also limits the interpretability of the results in this manuscript since the effect of bad FAR and POD, which lead to overestimation (high FAR) and underestimation (low POD), cannot be studied in the resulting rainfall fields. I strongly suggest to carry out the analysis of the rainfall fields for different subsetting variations. The most commonly used ones for comparing rainfall maps seem to be: 1. No threshold 2. Reference > 0.1 mm. This does not mean that all the plots have to be done several times, but at least the main skill metrics should be provided for the different subsets.**

*We agree that for the comparison with the previous studies, a set of indicators with the same (or similar) settings is needed. We computed continuous indicators with the filter set as Reference > 0.1 mm/h we added them to Table 2 and the sentence "To make easier the comparison with past works, we computed continuous indicators with the filter set as Reference > 0.1 mm and with no filtering at all. Results with the first setting yields slightly worse indicators, increasing the ME to -0.41 and the CV to 0.95, with a second digit increase of R2, around 0.5. The no-filter run shows values of ME and R2 in line with our original results, while CV is greatly affected by very small rainrates." in the reviewed manuscript.*

*For the categorical scores, instead, no filtering was ever performed: the threshold of 0.1 mm/h was used to discriminate wet and dry samples in the confusion matrix. We stated this explicitly in the revised document: "Thus we have set a wet-dry threshold equal to the minimum rain quantity detected by the tipping bucket rain gauge, i.e. 0.1 mm h$^{-1}$, for both estimate and reference". More information is provided answering the specific comment on L298.*

**2. Since a large part of the quantitative analysis is based on interpolated rainfall maps, I strongly suggest to show several examples of interpolated CML rainfall maps, e.g. of one or two specific events and e.g. accumulated over the whole period.**

*We welcome this suggestion, and we added a section (4.2.2 in the revised manuscript) where three case studies are reported, including the discussion of meteorological conditions, reporting interpolated maps and indicators. Moreover, in the Supplement, we discuss the cumulated map.*

**Minor general comments:**

- **Choice of POD and FAR: I assume (since it is not specified in the manuscript I looked at other papers that**
45  **use RAINLINK) that POD is hits/(hits+misses) and FAR is false_alarm/(hits+false_alarm). If this is the case, POD is the true positive rate (TPR). Wouldn't it than be better to use the false positive rate (FPR), like it is used in the ROC curve, instead of FAR. FPR and TPR are both normalized by the reference conditions. FAR instead is normalized by the predicted positive conditions. Can you elaborate on this choice?**

*The Referee understood correctly: POD = hits / (hits + misses) and FAR = false_alarms / (hits + false_alarms).*
50  *This choice to favour FAR over FPR derives from its common use in deterministic precipitation forecast/estimate validation (Tang et al., 2020; Petracca et al., 2018; Puca et al., 2013, McBride and Ebert, 1998, among others). The FPR= false_alarm/(false_alarm+correct_negatives), more common in probabilistic forecast verification, is heavily influenced by the most populated category (correct_negatives): in case of small scale, sparse or intermittent rain phenomena, FPR can decrease without any skill in the forecast since*
55  *the no-rain condition is the most common in the target area. FPR, for the same reason, could also be misleading when different seasons/climates, with different rain occurrence, are compared*

- **The writing needs improvement throughout the manuscript, in particular the introduction and conclusion. Hence, I stopped very early to note down technical corrections and suggestions for stylistic improvements when reading the manuscript.**

60  *We improved the revised manuscript through a careful review of the language.*

**Specific comments:**

**L28: It would be good to have another or additional reference for the claim that the "last generation polarimetric systems have only partially mitigated" the radar QPE problems. The book by Ryzhkov and**
65  **Zrnic, 2019 is certainly a very valuable textbook, but is is hard to find this conclusion in a 450 page reference. Access to it might also be limited.**

*We agree with the Referee, and we included few references more focused on the QPE of polarimetric radar performance evaluation: Figueras i Ventura et al., 2012; Gou et al., 2019.*

70  *Figueras i Ventura J, Boumahmoud A-A, Fradon B, Dupuy P, Tabary P. 2012. Long-term monitoring of French polarimetric radar data quality and evaluation of several polarimetric quantitative precipitation estimators in ideal conditions for operational implementation at C-band. Q. J. R. Meteorol. Soc. 138: 2212–2228. DOI:10.1002/qj.1934*

75  *Gou, Y.; Chen, H.; Zheng, J. Polarimetric Radar Signatures and Performance of Various Radar Rainfall Estimators during an Extreme Precipitation Event over the Thousand-Island Lake Area in Eastern China. Remote Sens. 2019, 11, 2335. https://doi.org/10.3390/rs11202335*

**L32: ": : :accuracy is still under evaluation (Tan et al., 2018): : :". This statement is a bit weak. In addition, there are many studies that evaluate the performance of IMERG, also on a broader level than Tan et al.,**
80  **2018.**

*Of course, many papers are dealing with satellite product validation, but very few of them deal with high resolution (hourly) data, mostly focusing on daily to annual integrals. We rewrote the sentence to be more precise:"... their accuracy is difficult to assess at high spatial and temporal scales (Tang et al., 2020)...", and included a more recent and pertinent reference*

85 *Tang, G., M. P. Clark, S. M. Papalexiou, Z. Ma, Y. Hong, Have satellite precipitation products improved over the last two decades? A comprehensive comparison of GPM IMERG with nine satellite and reanalysis datasets, Remote Sensing of Environment, Volume 240, 2020, 111697, https://doi.org/10.1016/j.rse.2020.111697.*

90 **L34: I think "broad diffusion" is the wrong term here. Something like "ubiquity" would fit better.**

*We agree and replaced "broad diffusion" with "ubiquity".*

**L36: "Accurate algorithms were introduced to measure : : : drop size distribution : : : water content". Since the sentence before talks about CMLs, the used references do not fit here, since they did not use, or only**
95 **partly used, CML data. Dual-frequency and dual-polarization data, as used in the references, is mostly not (yet) available in operational CML networks.**

*The Referee is right: the sentence was not correctly contextualized, and we reworded to: "Accurate experiments with high-quality links and numerical simulation were used to assess the capability of microwave links to measure average rainfall rates (Rahimi et al., 2003), drop size distribution (Rincon and*
100 *Lang, 2002; van Leth et al. 2020) and water content (Jameson, 1993). On the same token, the possibility to reconstruct a spatially continuous rainfall field relies on a sufficiently high density of the links, making the CML approach of particular interest for urban areas...".*

**L38: ": : :a spatially continuous rainfall path: : :" It is not clear to me what that means. Please rephrase.**
105 *The word "path" is replaced by "field".*

**L50: This is a very long and confusing sentence.**

*We rewrote the sentence to "Even if the general relationship between signal attenuation and rain rate is already well established, the successful use of CML data to quantitatively monitor precipitation still depends*
110 *on the quality and technical characteristics of the transmitted power data and the fine-tuning of the algorithms."*

**Section 2.1: What is the power quantization of P_min and P_max? Please specify.**

*We reported the quantization value (1dB) in the revised manuscript: "...are measured by the provider with*
115 *the resolution of 1dB at a frequency of ten..."*

**Fig1. and section 2.1.1: Are there pixels without a CML, i.e. LC = 0. This is not clear from the map and the text. Please clarify. If yes, what are the implications. E.g. if you would have to interpolate a rainfall field over two empty pixels in the west of Parma that would decrease the performance a lot compared to pixels that at least have one CML.**

*We changed the colour scale of Figure 1, to make clear the presence of few LC=0 grid boxes. Nevertheless, we do not think that cells with LC=0 represent an issue because we aim to evaluate an interpolated product whose goal is precisely filling the empty gaps between separate measurements. Previous CML papers also show rainfall maps interpolated at a finer scale (1 km) and with sparser and more inhomogeneous CML networks (e.g. Overeem et al., 2016). Besides, we agree that better results are likely to be expected from regions with higher coverage and we already addressed the matter throughout the analysis of the LC dependency in Section 4.2.1. Near the end of that Section, we added: "It seems that the sensitivity to LC could explain the improvement in the FAR of the RRB area, but not the sharp decline in the POD, suggesting that LC is probably not the only variable at play there. In Reno basin. These results integrate the findings of Overeem et al. (2016b), that highlighted the positive impact of higher LC on CV and CC at lower spatial resolution."*

**L170: It would be good to know what "set of consistency checks" has been used. Is everything done as in Overeem et al 2016? Even if yes, a short summary (2-3 sentences) would be good so that the reader does not have to go through the explanation in the reference.**

*The consistency criteria require that: the frequency is inside a specified range; there are no multiple occurrences for the same ID and DateTime, every ID always has the same geographical coordinates, not-available (NA) entries are not present. We added a sentence that clarified this point and added some statistics about the rejected data in the Supplement. Paragraph 3.1.1 now reads: "**1. Preprocessing:** the raw input goes through three consistency checks concerning data formatting and labelling. Any multiple observations for the same LinkID and DateTime are discarded, each LinkID is verified to maintain the same metadata throughout the whole dataset (Frequency, PathLength and antenna coordinates) and rows with NA values in any of the columns except for Polarization (which is supposed vertical if not indicated) are discarded as well.."*

**L174: What is "a comparable decrease"? Please be more specific.**

*Wet-Dry Classification is described in Appendix C of Overeem et al. 2016, and we used exactly their procedure. The description was here treated only qualitatively on purpose, but we now added quantitative references to increase clarity.*

*We modified the point 3.1.2 as follows:*

*"2. **Wet-Dry Classification**: the samples are discriminated in wet and dry periods by assuming that rainfall is correlated in space, through the so-called Nearby Links Approach (NLA), which works as follows. For each link, a time interval with a decrease in the received power is labelled as wet if at least half of the links in the vicinity (within 15 km radius) experience a comparable reduction, i.e. if the medians of the attenuation and the specific attenuation of the nearby links are below − 1.4 dB and − 0.7 dBkm-1 respectively. This is the second most computationally time-consuming step of the algorithm.*

**L178: Also here, it would be good to get more info on the outlier filters. What exactly was done? And even more important. How much data was removed?**

*We added some additional details on the procedure. Also, statistics on outliers will be added in the supplementary material.*

*We reworded the point 3.1.4 as follows:*

*"**4. Outliers filter and power correction:** outliers due to malfunctioning links can be removed again by assuming that rainfall is correlated in space. The filter discards a time interval of a link for which the cumulative difference between its specific attenuation and that of the surrounding links over the previous 24 h (including the current time interval) becomes lower than the outlier filter threshold, which is fixed at -32.5 dBkm-1h. After removing the outliers, the classification information is used to clean the receiving powers of the noise over the dry periods. The corrected powers $P_i^{Cor}$ will be equal to $P_{ref}$ on dry periods and $P_i$ on wet ones."*

**L186: Did you use specific a and b values from van Leth et al, 2018? If not, it is not clear why this is cited here. Please cite the source of the a and b values.**

*Van Leth et al. (2018) is cited here only to support the assertion about which variables the a and b parameters are sensitive to. To avoid misunderstanding, we removed the Van Leth et al. citation and completed the sentences mentioning the source of a and b: "are finally calculated using the k-R relationship $R = ak^b$ , where the coefficients a and b were from Leijnse 2007 and Lejinse et al 2010 for vertical and horizontal polarization respectively."*

> *Leijnse, H., 2007: Hydro-meteorological application of microwave links - Measurement of evaporation and precipitation. PhD thesis, Wageningen University, Wageningen. See page 65. Provided for frequencies from 1 - 100 GHz.*

> *Leijnse, H., R. Uijlenhoet, and A. Berne, 2010: Errors and uncertainties in microwave link rainfall estimation explored using drop size measurements and high-resolution radar data. J. Hydrometeorol., 11 (6), 1330–1344, doi:https://doi.org/10.1175/2010JHM1243.1.*

**L199: What was the length of the CMLs below 10 GHz? Even at 30 km (the maximal length in your data set according to section 2) a CML with 5 GHz is very insensitive to rainfall (approx. 0.05 dB at 1 mm/h path-averaged rainfall) so that light to moderate rain might not cause a detectable signal. Can you make sure that this does not have negative effects on the rainfall fields for light and moderate rainfall events (in the range 1-10 mm/h)? Couldn't it be that CMLs with zero rain rate are introduced in the interpolation method, which would better be left out? How much CMLs would you loose if you do not include CMLs below 10 GHz and how much does the "spatial coverage" decrease?**

*This is a very important point, and we thank the Referee for having it highlighted. In our network, we have only five links between 5 and 10 GHz. We will remove them, and we do not expect any major change in the results. A complete statistic on the CML characteristics is now presented in the Supplementary Material.*

*Moreover, we went deeper into this analysis, following also the comments of the Anonymous Referee #2: we investigated the sensitivity for all links, calculating the theoretical sensitivity through the inversion of the kR relationship at a fixed 1 mm/h rainrate. It has to be remembered here that the manipulations within the algorithm (especially the Aa threshold) do not allow a direct translation of the theoretical sensitivities into actual instrumental uncertainties or error bands.*

200 *The analysis is presented by means of the plot below, where all links are distributed according to their length (x-axis) and frequency (y-axis) and where the theoretical sensitivity field is showed as contour lines of equal sensitivity with small differences for the two polarizations.*

*We decided to remove from the dataset all the 15 links with a sensitivity below 0.1 dB per mm h $^{-1}$ (among which are the 5 low-frequency ones), here highlighted in red. We added the following sentence to the*
205 *revised manuscript: "The CMLs' operational frequency in our region spans between 5.0 and 45.0 GHz. We decided to extend the default frequency allowance window from 12.5 - 40.5 GHz (as was in the Netherlands) to 10.0 - 45.0 GHz, leaving out five low-frequency CMLs. We also removed from the dataset 10 other links with higher frequencies but with sensitivities below 0.1 dB per mm-1 (see Supplement for more details). This is done to avoid contamination by coarse low sensitivity signals."*

[Figure]

210

**L209: My feeling as a non-native speaker is that "delineate" is the wrong term here.**

*We replaced "delineate" with "detect".*

**Section 3.3: It is not clear from which reference you took which skill indicator. In my opinion, it would be**
215 **best to define the skill indicators here to avoid any misconceptions.**

*We added the description of the indicators to be more explicit, with the following sentence: "In the present work, we selected two sets of classical skill indicators, broadly used in the validation community (Nurmi, 2003): the first one is to assess the capability of the product to detect rainfall occurrence (categorical indicators), and the second one is to evaluate the skill in estimate correctly the quantitative precipitation*
220 *rate (continuous indicators). The first set is computed after a definition of a confusion matrix by counting the number of samples where both estimate and observation agree on classifying wet (hits, H), or dry (correct negatives, CN) samples, and where there are misses (M, observed wet and estimated dry) or false alarms (F, observed dry and estimated wet). Namely, Probability of Detection is defined as POD=H/(H+M ), the False*

*Alarm Ratio as FAR=F/(H+F), the Multiplicative Bias as MB=(H+F)/(H+M) and the Equitable Threat Score as ETS=(H−Hrnd )/(H+M+F), where Hrnd represents the number of hits obtained by chance. Given* ei *and* oi *as estimated and observed values resp., continuous indicators are the normalized Mean Error, defined as* ME = Σ(ei − oi)/ō, *the normalized Mean Absolute Error, defined as* MAE = Σ|ei − oi|/ō), *the Coefficient of Variation (CV) defined as the root mean square error divided by the mean of the observed values* ō, *and the Pearsons' Correlation Coefficient (CC), as the covariance of observed* oi *and estimated values* ei *divided by the product of the two standard deviations (Nurmi, 2003, Overeem et al., 2016b)."*

**L214: Complicated sentence and unclear formulation. I guess you are trying to say that your CML and reference products have a lot of rain rate that are so low that they can be neglected in any application.**

*The Referee understood correctly. However, we reworded the sentence to improve clarity: "Both interpolated CML and reference field have a large number of very low positive values (below 0.1 mm h$^1$ that do not have any physical relevance, but which are potentially very influential in normalized error metrics".*

**Fig. 2: It would be good to know the minimal distance from the individual CMLs to the reference rain gauge.**

*In the legend of the revised Figure 2 are now indicated the shortest distances between the link paths and the respective raingauges. The distance is always shorter than 3 km (well below any decorrelation length of precipitation in our region) and still under 70% of the respective link length (see figure below). Further inspection showed that there is no relation between the distance from the raingauge and the link performance. We added a sentence in the revised manuscript: "We have selected links in rural areas and different terrains with an active raingauge close to the link: the distance between link and raingauge, reported in Figure 2, is always below 3 km (significantly lower than the correlation distance of precipitation in Italy (Puca etal., 2014)) and always lower than the length of the link itself. In general, no dependence of the link performance on the distance from the raingauge is found."*

[Figure]

**Fig. 3: This figure contains a lot of useful information. It is a bit unstructured, though. It could be cleaned up by aligning the x-axis of each column and by sharing the legend in row 1 and 2. In row 3 two columns for "back" and "forward" could be used. Reusing the colors from row 2 in row 3 for different variables is also not ideal. The x-axis tick labels are also different in row 3 from row 1 and 2, so that it is not clear if**

**the depicted periods are exactly the same. Hence, in particular the alignment of the x-axis would help. If you redo the plot, which is what I would suggest, than you could also reconsider the order of the rows. I feel that starting with the raw data (now row 2) would make more sense since this follows the CML data processing workflow. The meaning of the pink horizontal band, explained in the text, should also be nna explained in the figure caption.**

*We re-designed Figure 3, following the Referee's advice, which was much welcome. We also removed the bright-band case following comments on L288 and on Section 4.1.3 by the Anonymous Referee #2.*

[Figure]

**L262: Can the overestimation of the one CML be explained by the spatial distance between this CML, the other CMLs and the gauge? If not, what is your explanation?**

*The spatial distances between the Sant'Agata links and gauge are all similar and very short. Many factors could concur in poor performances, but, unfortunately, we are not able, with the current reference dataset, to dig deeper into this issue. For example, different links orientations could lead to different wetting rates of the antenna's radomes in case of winds, or very local phenomena (as hails or showers) could perhaps hit one link and not the others nor the raingauge. Specifically, the mentioned overestimation seems generated by a slightly stronger peak in the middle of the event, common to both sublinks of the CML, located 1.85 km away from the raingauge. The three CML are so close together that the issue is probably wholly wiped out by the smoothing of the interpolation process. We added this comment: "cannot be certainly related to any macroscopic characteristics of the three links".*

**L269: Does this CML show these differences between P_max and P_min during the whole period? If yes, are there other CMLs that show something similar? Do you have any explanation or mitigation strategy? Regarding an operational application of CMLs there should be a way to deal with this kind of signals.**

*The gap between Pmin and Pmax is to our understanding a characteristic common to all CMLs and to the MinMax sampling strategy. It happens every time there is a variation of receiving power within the time*

280   *interval. If the gap persists in time, then there are probably some power fluctuations that must have a frequency higher than 15 min$^{-1}$ and an amplitude equal to the gap. This seems to us more a feature of the MinMax sampling strategy rather than an issue to be mitigated or dealt with. We added it more clearly in the revised manuscript: "Looking at Figure 3d, Pmin does show a decrease coupled to the missed rainfalls, but Pmax does not. This behaviour of Pmax is not an issue itself, as the NLA classification relies on Pmin only,*

285   *but it indicates that there are power fluctuations which happen faster than 15 min-1. Rapid fluctuations, in turn, suggest irregular and scattered precipitation patterns, that actually could be a factor that affects the correct classification, since NLA relies on the spatial correlation of the rain field. Therefore, a Pmax signal always near the baseline could be a precursor of local NLA issues."*

*Nevertheless, we briefly verified that, if rain is sensed from the algorithm, Pmin and Pmax are likely to be*
290   *distinct, while the opposite is not true (there are cases of Pmin-Pmax differences, sometimes very relevant, not associated to rain or outliers). There is also no particular correlation between the gap width in dB km$^{-1}$ and the estimated rain amount in mm.*

*Sensitivity analyses (see also answer to comment on L296 ) showed optimal values for alpha (the parameter which weights between retrievals from Pmin and Pmax ) very similar to the default one (0.28 vs 0.33).*

295   *The ATPC corrections (see the answer to comment on Section 2.1.2 by Anonymous Referee #2) mostly amplifies already existing Pmin-Pmax differences, which are moreover sensibly bigger than the correction itself (10-40 dB against 4-6 dB respectively). Lastly, the increased spread will not directly affect NLA, as the classification exploits only Pmin observations.*

300   **L281: If I understand correctly this data set is not part of the data set for the main analysis of the paper, correct? Please clarify in the text.**

*That is correct, the Referee understood correctly. See answer the following two comments for changes in the revised manuscript.*

305   **Section 4.1.3: It would be important to know the height of the antennas and the estimated height of the melting layer or zero-degree level. Form the fact that the data is from the month of March is cannot be concluded that the CML measured mixed-phase or solid precipitation.**

*The geographical height of the ground under the antenna is known to the authors (not the one of the pylons themselves) but confidentiality restrictions with the data provider prevent us from being more precise.*
310   *Independent instrumental measurements of the freezing level in the atmosphere are known too, and they were carefully taken into consideration before discussing the melting layer case, but we chose not to show them to avoid cluttering the Section with data not related to the main discussion. For similar reasons, we decided eventually to remove the melting layer case form the revised paper. See also the following answer to comment on L288.*

315

**L288: ": : âˇ Žbright band' in the radar reflectivity maps and is thus easily detected". If you have a dual-pol radar with a working hydrometeor classification, then yes, it can be detected. If not, than this is quite hard to do for smaller scale precipitation events and on short temporal durations. I suggest to add some more details to the explanation in the text.**

320 *We based our statements on polarimetric data observations and assessed the presence of bright band without any doubt. However, after the suggestion of the other Anonymous Referee also, we decided to drop this section about the melting layer entirely, being a little out of the main direction of the work.*

**L296: Is this underestimation due to missed event or to a general underestimation of the CML rain rates?**
325 **And why didn't you try to adjust the wet antennae compensation to compensate this underestimation? Overeem et al. 2013 and 2016 calibrated the wet-antenna compensation for a specific subset of their data, so it might neither be optimal nor applicable to your data. Please explain.**

*The first question could be addressed by comparing the overall values ME (-26%), indicating the relative deficit of measured rain amount, with the MB (0.77), the relative occurrence of estimated wet samples with*
330 *respect to the real number of wet samples. The underestimation seems to affect for a 23% the number of "events" and a little bit more (26%) the amount of water. From the conceptual point of view, however, the two things are tightly connected: the underestimation of the rainrate results in an underestimation of rain occurrence, as soon as the underestimate affects rainrate values just above the threshold.*

*As for the second issue, our feeling is that the reference data we considered (used in operational offices) are*
335 *not suitable to be used as a calibrator, in term of quality and spatial and temporal characteristics, as also the other Referee remarked. Anyway, we performed some trials with decreasing Aa, for the single-link analysis.*

*First, we checked for the 27 links with a close-by 15-min raingauge in which Aa and alpha values were producing the best overall performances, and the results are summarized in the following figures.*

[Figure]

340 *The CC surface (left) shows a clear maximum at: alpha = 0.3, Aa = 0.7, while CV (right) reaches no local minimum in the examined domain but has a plateau-like area of fair performance in which falls the best match for CC. This analysis suggests an Aa value much smaller than the one used by RAINLINK and in our work (2.3 dB), while the alpha parameter remains almost unchanged. However, looking at the PDF of the estimated rainfall emerges that the physical representativeness drops down with the lower Aa. Transferring*
345 *these results on the whole datasets also leads to worse overall results: CV worsens of +0.05 with no sensible improvement on $R^2$, the same for FAR and ETS, which change resp. of +0.08 and +0.01.*

*In our opinion, this indicates  that the major criticalities of the algorithm are to be found somewhere else (probably in the classification process or in the outliers filter), and that they should be addressed first, before fine-tuning these retrieval parameters.*

[Figure]

350

*We added a comment: "However, a simple sensitivity test, carried out to assess the impact of a decrement in the Aa value on the single link scores, did not show any substantial improvement, especially when its results are extended to the whole dataset. More information are provided in the supplementary material. "*

355 **L298: Since your study and the two other studies all use a different "Filter" (see your Table 2) the results are not really comparable. In particular your choice of "Ref. AND Product > 0.1 mm h⬜1" neglects the negative effects of false positives and false negatives. See my major comment above.**

*We agreed on this and run the calculations with the filter on the Reference only and without any filter too. As expected from the fairly good categorical scores (which evaluates all the four values of the confusion*
360 *matrix), false positives and negatives do not affect the overall picture too much.  We added specific columns to Table 2 to allow comparisons with other studies.*

**L318: "The accuracy in the estimates is reached at the expense of POD, ETS and BIAS: around 50% of the rainfall duration is lost in this area". I understand that when FAR is lower (mentioned in the sentence before) and POD is lower there are less rain events, both correct and incorrect ones, in the resulting CML**
365 **rainfall time series. That would explain that there is even more tendency to underestimate here. But, if I understood correctly, the bias is only calculated from values where both CML and reference are above 0.1 mm/h, so that false and missed CML rain events have no impact on the calculation of the bias. Can you elaborate on that?**

*First, we made a mistake: "BIAS" (undefined in this work) stands for Multiplicative Bias (MB), i.e.*
370 *number_of_estimated_wet)/(number_of_observed_wet). Hence a Multiplicative Bias of 0.47 indicates that only half of the wet samples are found. Then, we have to remind that categorical scores in our work are always calculated over the unfiltered dataset, around the threshold, which would later be used in the filter. This was not sufficiently clear, and we already presented the changes in answer to the first general comment and the one on L298.*

375     *Addressing now this case specifically, the amount of rain lost in this area (given by ME) is similar to other areas, and the indicators of numerical accuracy of the estimates (CV and CC), are quite high. This indicates that in this area, the rainrate is estimated with higher accuracy, while the discrimination wet/dry is worse.*

    *We modified the paragraph to: "The higher accuracy in the estimates is reached at the expense of POD, ETS and MB: around 50% of the rainfall duration is lost in this area. The main peculiarity of the RRB area is the*
380     *high LC, which is 50% higher than the rest of the regions. We can infer that the higher coverage led to a more selective NLA classification, which reduced FAR and POD. The marked improvement of continuous indicators suggests that the quantitative matching between estimated and reference could be positively related to LC."*

385     **L324: Remove the "For" at the beginning of the sentence**

    *Ok, thanks.*

    **L327: ": : :this suggests that LC is probably not the only variable at play there". This is good to know, since that would have meant that regions with high CML density perform bad with the used algorithm. The**
390     **CML data set of Overeem et al 2016 also has regions with a very dense network and regions with a coarser network. Hence, a strong dependence of the RAINLINK algorithm on LC should have already been noticed by them. Could it be that there is one CML in this area that shows "strange" behavior, e.g.strong fluctuations, that negatively affects the POD of the many surrounding CMLs by not letting RAINLINK do the detection of rain events?**

395     *Overeem et al. (2016) showed how the CML performance varies against the mean link density (our "LC") by analysing normalized variance and correlation on a 74km $^2$ grid. Our results for CV and CC estimated at 25 km$^2$ are in good agreement with them. In addition to their work, we also show the effect of LC on the categorical indicators, providing some interesting results for the FAR, especially and giving more insight into the topic in general. However, we were not able to isolate all the sources of uncertainties and to gauge the*
400     *performances of the single links individually. We added a sentence in the revised manuscript: "These results integrate the findings of Overeem et al. (2016), that highlighted the positive impact of higher LC on CV and CC at lower spatial resolution".*

    **L355: Since your reference data set ERG5 is an interpolated rain gauge product, it might miss small scale**
405     **rain events compared to the radar. Assuming that clutter removal was done in a sufficiently good way, the radar should not have a high FAR in general. Couldn't the fact that ERG5 might miss some real rain events explain the high FAR of the radar product?**

    *The clutter is removed through a static map of clutter, a beam trajectory simulation, and an anomalous propagation cancellation (see Fornasiero et al., 2006). Moreover, WiFi/WiMax signals are filtered through a*
410     *decision tree and fuzzy logic techniques which exploit Z, Zdr, W, V, and Z and Zdr variance. We do not think, therefore, that the clutter is the reason for the high FAR. We suppose instead, as the Referee pointed out, that a reason for high false alarms ratio could be that ERG5 misses some small scale events. We modified the sentence: "...while rain gauge (as well as the reference product ERG5) and CML networks...".*

**L362: ": : :making CML a more robust sensor." Robust in what sense? Please explain in more detail in the text.**

*We want to emphasize that to fully exploit radar capabilities a customized Z-R relationship should be used for each type of precipitation. At the same time, the k-R relation of the CML retrieval is almost independent on the DSD, due to its linearity in this frequency range (Leijnse et al., 2008). We added few words to the sentence: "thus making CML a more robust sensor, in the sense that the same coefficients for the retrieval can be effectively applied regardless the type of precipitation".*

**L365: When speaking about the "operational context" and the advantages of CMLs it should be discussed how the low POD, found in this study, affects the CML's potential for operational applications. This should be part of this paragraph.**

*We'll specify better the possible role of CML in the operational context, modifying the sentence to: "In an operational context, where several precipitation products (each one with its proper error structure) are available to the forecaster, it is of high relevance also their latency, i.e. the time taken from the acquisition of the primary data (the occurrence of the event) and the delivery of the product in a ready-to-use form..."*

*If the error structure of the products is well assessed, and the operator is well aware of it, he can decide to use a product that usually underestimates (low POD, as CML in our case), or a product that tends to overestimate (Radar, in our examples). At the end of the paragraph, we added the comment: "It is to the operators' preference, based on product error structure, current meteorological conditions, and user's requirements, to make use of the most suitable product".*

**Response to Anonymous Referee #2**

**Main weakness of the manuscript is related to CML data processing and data analysis which is based on open-source package RAINLINK applied on CML data in northern Italy. I am missing the definition and answering the important research questions which can provide new insights in CML rainfall retrieval. The overall scientific significance of the manuscript is fair.**

**Therefore, the manuscript needs major revisions. I see several aspects that can be studied using such data set. The quality of CML product is questionable and it show systematic underestimation. Then one way could be to test/develop other processing methods of CML data to reduce this bias and improve the quality of the product. Other interesting point could be an orographic aspect which is mentioned in the manuscript, but not studied in detail.**

*We agree with the Referee that in this paper we do not address fundamental research questions, such as to set-up advanced algorithms or tackle challenging issues. Still, we think that one important task in the research activity is to communicate to potential users possible applications of the research itself.*

*Moreover, in our opinion, the data available to us was simply not accurate and complete enough to develop and test new algorithms or to analyse the impact of orography or other known critical aspects of the rain retrieval from CMLs. Longer data time series over wider regions and a more reliable and representative reference dataset is needed to do such studies, which was not accessible to us at the time. However, we believe that this work demonstrates a good potential of the technology even at its most basic implementation, and gives valuable hints for future regional improvements.*

*The objective of our study, indeed, was to test the possible role of CML retrievals in an operational environment without any previous research on the characteristics of the available CMLs. We performed an "out of the box" approach as we aimed to test the performances obtainable without specific calibration (whose related effort could be not sustainable in many places). We assessed that a robust and freely available algorithm (such as RAINLINK) provides a product with spatial and temporal characteristics comparable to products routinely available to the operators in our region. Moreover, we highlighted how the performances of RAINLINK could be improved, addressing the few parameters that could benefit from a calibration/validation campaign (with proper instruments), once it will become possible.*

*To clarify, we'll modify the sentences in the revised Introduction as follows:*

*".The first objective of the present work is to make a validation of precipitation amounts and distributions estimated only from CML attenuation data, using a well-established, freely-available algorithm (i.e. RAINLINK), over two areas of interest in the Po Valley (provinces of Bologna and Parma), where CML data have been obtained from Vodafone (direct purchase). Both areas contain river basins of considerable local interest, which will be explicitly addressed. Moreover, we consider for intercomparison only precipitation products routinely available at Meteorological Service of the Regional Agency for Environmental Protection and Energy (Arpae-SIMC): this, from one side, prevents us from performing a proper calibration of the algorithm, but, on the other hand, allows us to understand how CML product behaves with respect to other operational prod- ucts. The further aim of the validation study is thus to test the potential of the technology even at its most basic implementation, indicating where to direct the tuning efforts, to set the background for possible inclusion of CML data in the operational routine procedures for precipitation monitoring."*

**General comments:**

**1. The results show systematic underestimation of QPE derived from CMLs. RAINLINK package contains several strong assumptions (constant WAA of 2.3 dB, constant k-R parameters etc. ) which can influence the results significantly. Recent knowledge shows that WAA is complex process with many unknowns (e.g. Leth et al., 2018). The dataset probably contains a certain portion of sensors with low sensitivity (this is reviewer assumption since the CML statistic is not provided) to rainfall where WAA can play dominant role in resulting rainfall retrieval. I would recommend to make at least sensitivity analysis of the results to most significant parameters.**

*It is well known (van Leth et al., 2018 among many others) that antenna wetting is one of the main problems in microwave estimation of precipitation, and for this reason, we think we cannot address this issue with our operationally oriented verification system. We remark that to address this issue properly, van Leth et al. (2018) deployed a unique experimental setting, with extremely controlled antenna conditions (time-lapse camera pointing the antennas) and accurate reference measurements (five disdrometers along link path). Even with this unique experimental setting, neither van Leth et al. (2018) could definitively address this issue in a general way.*

*Anyway, to give some more hints to the reader interested in the use of RAINLINK, we performed a sensitivity study to test the impact of the Aa (Antenna attenuation) parameter on the results, though bearing in mind that Overeem et al. (2016) itself reports that " applying Aa should be seen as a pragmatic approach towards correcting for wet antennas". The study is performed on the sample dataset of the 27 CML against the nearby raingauges, exploring the sensitivity to Aa and alpha parameters paired and using CC and CV as loss functions. The results are summarized in response to the Anonymous Referee #1 about L296 and added to the Supplementary Material.*

*For the discussion about the dataset characteristics and the low sensitivity links, please refer to the reply to the specific comment on L90-94 instead.*

500

**2. Spatial interpolation is based on assumption the path-integrated rainfall is represented as a point measurement. This assumption can be used for rough grid 5x5 km and shorter CMLs. However, it is weak for single link comparison (section 4.1) including single event comparison. Here, spatial-temporal structure of rain together with the layout of given RG and CMLs can play significant role. Then it is**
505 **impossible to compare single point measurements and CMLs observations.**

*We thank the Referee for pointing out this issue, and we agree that the comparison between single link and single rain gauge is affected by many uncertainties. However, similar shortcomings should also apply when comparisons with other instruments are considered. The only way to proceed correctly seems to be to follow Van Leth et al. (2020)'s approach or similar (i.e. many instruments along the link path), which unfortunately*
510 *is impossible to replicate in an operational scenario.*

*Still, we believe that our single-link vs raingauge analysis gives the reader some valuable hints to understand the behaviour of the interpolated RAINLINK product when presented.*

*To corroborate the robustness of the analysis, we added in the new version of Figures 2 and 3 the information about the shortest distance between the CML and the respective raingauge. These distances are*
515 *distributed as discussed in answer to the comment on Fig. 2 by the Anonymous Referee #1, who raised a similar question, and it can be seen that a sufficient level of consistency is always guaranteed.*

*We, therefore, added the following sentence in the new manuscript: "We have selected links in rural areas and different terrain with an active rain gauge close to the link: the distance between link and raingauge, reported in Figure 2, is always below 3 km (significantly lower than the correlation distance of precipitation*
520 *in Italy (Puca et al., 2014)). In general, no dependence of the link performance on the distance from the raingauge is found."*

*Basing on the various studies from the literature where space integrated rainfall is compared with raingauges measurements, we are not convinced by the last sentence of the comment.*

525 **3. Since rainfall maps are the key product of the presented study, I would expect to show visually CML rainfall maps – event-based or cumulative rainfall compared to reference.**

*According to also the hints of the Anonymous Referee #1, we added three case studies in the revised manuscript, with maps and skill indicators. We will show also the cumulated map for the whole period in the supplementary material.*

530

**4. I am missing relevant discussion section in the paper**

*We would prefer not to change the structure of the paper, but we added deeper discussion in many parts of the manuscript. Moreover, we added a new chapter, supplementary material, and completely rewrote the conclusions, where more discussion is reported.*

535 **5. I am not satisfied with the conclusions which do not provide novel information beyond the state of the art in the field of CML rainfall retrieval.**

*We better specified in the Conclusions that the focus of the paper remains on the operational implementation of the CML-based rain retrievals: the novelty, therefore, consists in evaluating the RAINLINK performance in a situation different from the original one, exploiting different metrics, techniques, and operational products as benchmarks, gauging in the process also the implementation effort and indicating the principal strength and weaknesses while suggesting ways to benefit from CML products the most. We completely rewrote the Conclusions also to include comments on the newly presented event-scale analysis. Now the conclusions reads as:*

*"An assessment of the 445 rainfall retrieval capability of CML opportunistic sensors over heterogeneous terrain in northern Italy is conducted at different spatial and temporal scales for two months of data. We implemented the open source RAINLINK algorithm in a new area and context, where no regional CML studies had previously been performed. We evaluated its performance through a complete validation scheme which involves operational precipitation products as benchmark, gauging in the process also the implementation effort and identifying major strengths and weaknesses to make a profitable use of CML products.*

*First, 26 CMLs (out of the total 308) are compared with the closest raingauges at a 15 min scale. Overestimation and underestimation of rain amount are both present, though the latter appears dominant. A marked variability among different links does not prevent to achieve a generally acceptable skill (CC from 0.50 to 0.88). The wet-dry classification approach and the value of the wet antenna correction may generate loss of rain amount in case of small scale and/or intermittent episodes.*

*Finally, higher elevation CMLs show in general worse performances. Interpolated products obtained from the full sample of 308 links confirm that a non-negligible quantity of rain is missed (normalized Mean Error is -0.26, overall CC is 0.68 and overall CV is 0.78), but also show that the rain retrieval capability is suitable for operational application, especially if the product is integrated over large areas (CC rises to 0.92). Higher link densities increase the quality of the CML estimates at both gridbox and basin scales, mostly in terms of decreased FAR.*

*Performances at event scale show enhanced skill in case of heavy precipitation, even in case of small scale rain episodes, while problems arise when light/moderate rainrates challenge the algorithm in the ways we already identified in the single-link analysis. Negative impact on the overall results comes from areas with poor sensor coverage, especially near the border of the areas, but it should be considered that also reference rainfall fields can be affected by shortcomings of the same nature.*

*Furthermore, when compared to other products currently available for operational real-time exploitation, CML sensors show similar or better abilities than their counterparts, especially if latency is also taken into account. Hence an integration of microwave links sensors in an operational service is highly desirable, even without a proper calibration of the algorithm to the local climatology and CML network characteristics.*

*When a more complete dataset would become available the validation scheme implemented for this work could be promptly used to tune the RAINLINK parameters (NLA radius, Aa, ) on a training sample specific of the study area."*

**Specific comments:**

**L. 33-45. I don0t agree with this paragraph since the first sentence refer to CMLs. The provide references are partly based on experimental microwave link setup, not CMLs. I wonder we know accurate algorithms for DSD, water content etc. based on CML observation.**

*The Referee is right: the sentence was poorly structured. We rewrote as: "Accurate experiments with high-quality links and numerical simulation were used to assess the capability of microwave links to measure average rainfall rates (Rahimi et al., 2003), drop size distribution (Rincon and Lang, 2002; van Leth et al. 2020) and water content (Jameson, 1993). On the same token, the possibility to have a spatially continuous rainfall field depends on the density and distribution of the links, making the CML approach of particular interest for urban areas..."*

**L. 70. observation period – since later in the manuscript some analysis are event based I would add into the Supplementary material information and data about precipitation events during observation period. For selected rainfalls and locations used later in section 4.1 some detailed rainfall metrics would be welcome.**

*We added Supplementary material where a section with a description of rainfall characteristics is presented. Moreover, in the new section 4.2.2, three case studies are shown, also discussing specific precipitation features.*

**L. 90-94 The usage of CMLs with low operating frequencies 6 – 15 GHz is questionable for QPE because of low sensitivity of those devices to rainfall even with longer path lengths. It would be useful to provide statistic evidence of different frequency bands in the data set including calculated theoretical sensitivity to rainfall. Then the effect of constant WAA to the results would be much clearer.**

*We thank the Referee for having pointed out this topic. In our network, we have only five links between 5 and 10 GHz, and a few more around 12 GHz. We investigated the sensitivity for all links, as presented in the answer to comment on L199 by the Anonymous Referee #1. Following that analysis, we decided to remove from the dataset the 15 links with a sensitivity below 0.1 dB per mm h$^{-1}$ and to add a sentence to the revised manuscript: "The CMLs' operational frequency in our region spans between 5.0 and 45.0 GHz. We decided to extend the default frequency allowance window from 12.5 - 40.5 GHz (as was in the Netherlands) to 10.0 - 45.0 GHz, leaving out five low frequency CMLs, but also to remove from the dataset 10 other links with higher frequencies but with sensitivities below 0.1 dB per mm h$^{-1}$. This is done to avoid contamination by coarse low sensitivity signals."*

**L. 104 Spatial distribution of LC – could you explain why the LC is lower in the main regional cities (Parma and Bologna) than in countryside – Figure 1?**

*In Italy and generally in the world, most of the CMLs in urban areas are being substituted by underground optical fibre cables. See also the reply to the comments for L228.*

**Section 2.1.2 Transmitting power levels I found this paragraph a little bit confusing. I would ask to rephrase it to provide clear information about ATCP processing.**

*We rephrased the paragraph as follows, describing the manual correction of the ATPC in detail and hopefully improving clarity and exhaustiveness.*

*"CMLs are usually equipped with Automatic Transmit Power Control devices (ATPC) which modulate the transmit power to guarantee a constant power level at the receiving end of the link, cancelling minor*

*fluctuations of the total attenuation along the path. ATPC works at a higher frequency than 15 min⁻1 and in*
620    *a power window spanning from 0 to +6 dB. With ATPC active, attenuation measurements should, therefore, be performed subtracting receiving to transmitting powers and are not possible from receiving powers only. The CMLs analysed in this work are equipped with ATPC, but we do not have access to the transmitting powers, due to confidentiality restrictions. Luckily, provider engineers gave us instead some statistics of the functioning of the ATPC devices (specifically, the modulation maxima (in dB) in the time interval), through*
625    *which we are able to correct the receiving power levels, compensating for the power modulation effects, simulating CML data with constant transmitting powers and thus allowing RAINLINK to estimate attenuations from receiving powers only. The correction intervenes on minimum received powers (Pmin), which are with no doubt affected by the ATPC: they are manually lowered by the maximum ATPC modulation applied within the respective 15 min time window. Maximum receiving powers (Pmax) instead*
630    *are left untouched as the ATPC working frequency and the 15 min⁻1 sampling frequency does not coincide and there was no way to infer a reasonable compensation. This could result in a broader gap between Pmin and Pmax.. This could result in the broader gap between Pmin and Pmax."*

*Implications of the ATPC corrections on the Pmin-Pmax gap are discussed in the answer to the comment about L269 by Anonymous Referee #1.*

635

**L. 193 Interpolation – please explain how path-averaged rainfall depth from each CML is implemented into spatial interpolation. This not very clear from provided description Section 4.1 Single link verification – see my general comment about point and pathaveraged rainfall estimates. This is difficult to understand especially when we don't see detailed information about precipitation metrics during**
640    **observation period. The data also does not correspond with previous statement in Section 2, that in higher altitudes are higher amount of rains.**

*We modified the description of interpolation procedure that now reads: "CML path averaged precipitation estimates are assigned to the mid points of the links like point measurements ("virtual raingauges"). Interpolation of the point-like measurements is performed at hourly scale with ordinary kriging on a*
645    *spherical semivariogram on the ERG5 grid. Sill and Range parameters are estimated from the available raingauge stations of three consecutive years. The interpolated field is truncated if it gets smaller than 0.05 mm, which is half of the minimum detectable rain from a raingauge."*

*More details on the precipitation characteristics involved in the analyses are now provided in the Supplementary Material.*

650    *The sentence in Section 2 is related to the rainfall climatology of the region and states only that the highest rainfall amounts are located on the hills. No direct correlation or proportionality between rain amount and height is suggested. From the additional information about the total cumulated rain depth over the two months, now provided in the supplementary material, it is possible to verify that the precipitation maxima are indeed on the meridional border of the region, which coincides with the Apennines ridge (see Fig 1 of the*
655    *manuscript).*

*To improve clarity, the sentence in Section 2 now reads:*

*"…, with the maxima of the rainfall amounts located on the Apennines ridge (see Supplement)."*

**L. 228 I suggest this statement as weak and confusing "They have been chosen in areas with different**
660    **terrain and network density and far from the cities, as CMLs in urban areas are already well studied and**

**also the most eligible to be replaced by optic fibres.“ I don0t see why CMLs in cities should work in different way than in country side. Is there evidence that CML in cities are already well studied and in the countryside not? Network development is not relevant for this paper and this sentence is speculation.**

665 *CML network characteristics are different going from cities to the countryside. Specifically, in the cities, there are fewer CMLs since most of them have been already replaced with optical fiber (see also the answer to the comment on L104 and the recent The Netherlands' situation reported for example in Overeem et al. (2016), Introduction section, 4th paragraph). We provide many references for metropolitan CML studies with short links. Moreover, implications of the network's developments on operational retrieval capabilities are among the most relevant topics in the CML field.*

670 *We removed the sentence and modified the previous sentence to: "We have selected links in rural areas and different terrains, that have an active rain gauge close to the link..."*

**Sections 4.1.1.-4.1.2. - Best and Worst Case Example – I do not understand why there is no text information and results interpretation with respect to rainfall intensity and rainfall characteristics. 4.1.2** 675 **represents light rain when the sensitivity to rainfall of CMLs is low. WAA is significant here anyway. Also, data provided from NMS system in form Pmin Pmax are limiting factor. This shows clear limits of CML for light rainfalls and Pmin Pmax approach.**

*We thank the Referee for this comment. As also suggested in the comment on L70, we added some Supplementary Material to the revised manuscript in which we describe the precipitation characteristics. We* 680 *added a more detailed discussion on these results, and we included three new event-scale validations, addressing primarily the type of precipitation and the Pmax-Pmin approach, and also considering previous Referee's comments.*

*However, regarding the case treated in the Section 4.1.2 specifically, we would first like to point out that the information about the precipitation characteristic for the duration of the event is provided in Fig.3 in the* 685 *form of 15min raingauge measurements (see Fig.3, solid circles and black cumulated profiles). Secondly, the links over Vergato have an average theoretical sensitivity of around 0.3 dB/mm/h, which is roughly the same as the links over Sant'Agata, so the theoretical sensitivity alone could not explain the different performances and behaviours between the two case studies. Nor could the wet antenna attenuation parameter Aa, as it is clear from the figure that the issues there are mainly related to the classification, which* 690 *misses most of the wet intervals.*

*Lastly, we do not see how the Pmin Pmax sampling should be a limiting factor here, as there are no evident anomalies within the presented signals. The fact that Pmax rarely moves from the baseline value, as discussed in the response to comment on L269 by Anonymous Referee #1, has no direct implications on the the classification, as NLA does not consider Pmax in its calculations. As stated in the original manuscript,* 695 *Pmax near the baseline indicates only that the power fluctuations are faster than 15 min $^{-1}$, which suggests irregular and scattered precipitation patterns. It is this irregular precipitation, and not the constancy of Pmax itself, that could be a factor that affects the correct classification, since the NLA algorithm is based on the spatial correlation of the rain field.*

*We modified the sentences to be more clear on the topic:*

700 *"Looking at Figure 3d, Pmin does show a decrease coupled to the missed rainfalls, but Pmax does not. This behaviour of Pmax is not an issue itself, as the NLA classification relies on Pmin only, but it indicates that there are power fluctuations which happen faster than 15 min-1. Rapid fluctuations, in turn, suggest*

*irregular and scattered precipitation patterns, that actually could be a factor that affects the correct classification, since NLA relies on the spatial correlation of the rain field. Therefore, a Pmax signal always near the baseline could be a precursor of local NLA issues."*

**Section 4.1.3 – I do not think that this melting layer story fits to this story. First, the data set is presented as spring – summer period. The article is focused on liquid precipitation, this is another story.**

*The melting layer episode does not belong to the 2-month dataset used for the main study, but it was a standalone dataset obtained from Vodafone for preliminary checking. This event occurred in early March when the freezing level could reach the ground, especially on the hills. Since liquid precipitation at midlatitude originates from frozen hydrometeors, the bright band is a rather common feature in our regions and introduces errors in the radar estimates often difficult to correct. Anyway, we understand that this issue is a bit far from the mainline of the work, so we decided to delete this subsection.*

**L. 320-330 I do not fully agree with those statements about LC. Different LC often means different frequency bands distribution. In the region with high LC one can expect higher frequency bands with higher rainfall sensitivity.**

*This would be true if LC measured the density of the antennas, but since it is an estimate of the cumulated link path lengths crossing a square of 5x5 km, LC also benefits of long links overpasses, which work at lower frequencies. In fact, we did not find any correlation between frequency and LC to date, but we thank the Referee for the hint.*

**Figures general – I found inconsistency when using brackets for units – none, () or [] in different figures**

*Thank you for noticing this, we have fixed this inconsistency.*

**Figure 7. I do not understand the "bad" results of adjusted radar in comparison to the reference which was used for radar adjustment. The results are comparable to unadjusted radar data. Could you explain that?**

*The adjustment is performed with gauges and not with the interpolated reference, as specified in Section 2.2.3. The procedure matches the rainrates estimated over the gauge locations but does not ensure the consistency of the whole radar field with the gauge interpolated one, mostly because of the high spatial variance of the radar field (as already discussed in Section 4.2.2, L356).*

*Therefore, discrepancies in the areal averages are not only to be tolerated but also expected. Moreover, the spatial autocorrelation of the G/R adjustment factor is even lower during convective events, leading to a less effective correction. The following figure shows an independent analysis of the radar adjustment performances in the same two months of 2016 and it is visible that around the low thresholds used in our paper (0.1 mm) practically no improvement is expected to come from the adjustment procedure.*

[revised manuscript text omitted]

---

## Author Response (AR2)

**Review of the first revised version of the manuscript "Commercial Microwave Links as a tool for operational rainfall monitoring in Northern Italy" submitted to AMTD by Roversi et al. in 2020.**

**Anonymous Referee #1**

**Authors response**

*We deeply thank Anonymous Referee #1 for the comments on the revised version, which we took in serious considerations and followed at our best. We provide a point-by-point response to the specific comments below and we attach a marked-up version of the second revision of the manuscript, which now has a separate Discussion section and a more exhaustive Section 3.2. We will also attach the previous revised version with the corrected visualization of the track changes.*

The authors did a thorough revision and provided good responses to my comments. They altered and extended their analysis accordingly. I still have some general comments that should be addressed which will require a minor revision, in the sense that I expect mostly only changes to the text and not to the underlying analysis or plots. In particular, the quality of the derived CML rainfall fields and their limitations are not communicated and discussed adequately.

I want to note that, in contrast to the initial assessment of reviewer 2, I think that a further extension of the presented analysis regarding an adjustment or extension of RAINLINK is not required, in particular after this first revision. Based on the limitations of RAINLINK that have been found in the presented analysis, it will, however, definitely be an important task for the authors (or for another research group with a large enough CML data set) to further investigate the transferability of RAINLINK and the need for its recalibration. But, in my opinion, this would go beyond the scope of this manuscript, which already provides enough new insights, taking RAINLINK to a new region with a new data set and doing a detailed analysis of its performance. It would however be important that the limitations of RAINLINK, in its unadjusted version as it was used in this work, are communicated clearer in the next revision of this manuscript (see my general comment on this below).

**General comments**

1. Insufficient discussion of limitations of RAINLINK:

   Since the authors only apply the existing RAINLINK algorithm without any adjustments or extension, one of the main contribution from this manuscript is to discuss its limitation and clearly state them.

   Fig 3. for the Vegato case shows a clear limitation of NLA (at least this is suggested in the text, see also my specific comments below). Fig 4 May 11 (middle column) also seems to support that.

   POD is quite low. ME shows clear underestimation and seems to be affected by POD (see my specific comment below)

   How are these metrics affected by parameters of RAINLINK? Is there room for improvement with adjustments of RAINLINK or better algorithms? Or is this also limited by the CML data? This should be discussed.

   The authors mention several of these issues at different places in the text, but, the information is spread across the manuscript. Since, in my opinion, communicating the limitations is important, I strongly suggest to have a dedicated section for this discussion.

   One sentence about this should also be added to the abstract.

   *We wrote a specific Discussion section where all the comments to the results are now gathered together and where the limitations of the "out of the box" approach are clearly and thoroughly stated. We also explained with more detail (and, we hope, clarity) the reasons behind the choice to follow this kind of approach and briefly what alternatives there could be. These new informations now integrate the Section 3.2 about the algorithm's implementation in Northern Italy. We*

*modified Abstract and Conclusions accordingly.*

2. Interpretation of results:

I do not agree with some of the interpretation on the quality of the results.

As written in my comment above, POD and ME indicate limitations of the derived RAINLINK CML rainfall products.

Stating that the performance of the derived CML rainfall product is "very similar, when not even better" (L11) to "adjusted radar-based precipitation gridded products" is a bit far-fetched in my opinion. The interpolated reference data set might be a good choice to compare the interpolated CML rainfall fields with, but I wouldn't say that the radar is worse just because it differs from interpolated gauge rainfall fields. The radar will detect small scale rainfall that the interpolated gauge data set just does not observe correctly or not at all.

Further examples on interpretations that should be more modest are e.g. in L376 and L379 (see my specific comments).

*We clarified that "when not even better" refers only to satellite products and we moved on to less optimistic statements throughout the manuscript. Nevertheless, we would like to stress the fact that the choice of using the gauges as "ground truth" resembles the operational and validation methods of the local weather service. The general principle that the radar could be capable of collecting small scale rainfalls is now stated more clearly in the manuscript.*

3. Strange PDF diff format. The blue text in the diff text has these unnecessary two lines of dots underneath the beginning of each word which makes it hard to read. If possible, for the next iteration, a more readable highlighting of added text could be used.

*We are very sorry for that. To our understanding it was an unnoticed issue with the software that merged the author response and the marked up manuscript together. The original diff pdf is much more readable and will be included below.*

4. Writing could still be improved. I add some examples as specific comments below, but I won't provide a complete list. Also note that I am not a native speaker, only somebody how typically gets "good grades" for the writing in papers. So this is a bit subjective.

*We thank the Anonymous Referee for the careful remarks. We performed another accurate revision of the language, hopefully reaching an overall sufficient level of proficiency.*

**Specific comments on the revised manuscript**

L7: "The results of the 15 min single-link validation with close-by raingauges show high variability, with the influence of the area physiography and precipitation patterns and the impact of some known issues". This is a good example for a sentence that, in my opinion, needs improvement. Instead of "raingauges" write "rain gauges" (based online English dicts), which should be changed throughout the manuscript. The added "the" before "influence. . . " is, in my opinion not correct. The formulation ". . . and the impact of some know issues" is too vague. I am also not 100% sure if I understand the sentence correctly. Maybe writing ". . . high variability, which can be explained by the area physiography. . . ". But, as I said, I am not 100% sure what is meant here.

*"Rain gauge" is now spelled correctly throughout the whole manuscript. We corrected the sentence as suggested, since the meaning was correctly understood by the Referee. The revised sentence reads:*

*"The results of the 15 min single-link validation with close-by rain gauges show high variability, which can be caused by the complex area physiography and precipitation patterns. Known sources of errors (e.g. the attenuation caused by the wetting of the antennas or random fluctuations in the baseline) are particularly hard to mitigate in these conditions without a specific*

*calibration, which has not been implemented. "*

85

L13: What is a "diffuse underestimation"?
*"Widespread underestimation" is probably a better formulation.*

L42: I would write something like "dedicated hardware" instead of "high-quality links". CMLa are also high-quality, just
90  with different priorities. *Corrected, thanks.*

L44: I have never heard the expression "On the same token". Online dictionaries say that the correct version would be "By the same token", but there might also be more commonly used expression that can be used here. *Removed, since it was not essential.*

95  L48: in my opinion it should be "...of the CML approach..." *Ok.*

L70: Write "on the one hand" instead of "from one side" since later in this sentence you write "on the other hand". *Ok, thanks.*

L71: "...prevents us from performing a proper calibration of the algorithm...". I do not understand how this prevents the
100  calibration. You wrote this also in the response to my comment to L296 and in response to reviewer 2. Are you saying that the products currently available are not reliable enough? Please explain?
*We have now made our reasons clearer adding various informations to Section 3.2: "As suggested by their authors (Overeem et al., 2016a), a solid calibration of the RAINLINK retrieval algorithm should be implemented exploiting numerous instruments along the link paths and organizing dedicated measurement campaigns, which were not feasible for us. The overall temporal*
105  *span should also allow the dataset to be split into two non-overlapping data sets for calibration and validation, but the to- tal wet hours available to us were not enough to grant a statistical significance for both sub sets. The gauge-adjusted radar product (which is commonly exploited in most CML studies) is not the one currently selected by the regional weather agency Arpae-SIMC as their quantitative reference, a choice that went in favour of the interpolated rain gauges product ERG5 (see Section 2.2.2). The spatial and temporal resolution of ERG5, however, is too low to perform an effective calibration. Therefore*
110  *we analysed some CML - rain gauge pairs only where the gauges were already in the vicinity of the links (Section 4.1), while we validated the rest of the dataset against the reference only through its interpolated product (Section 4.2.1 and 4.2.3)".*

Section 2.1.2: The revised manuscript contains the following formulation, "provider engineers gave us instead some statistics of the functioning of the ATPC devices (specically, the transmitting levels, thus modulation maxima (in dB) in the time interval),
115  ...". What does that mean? Were you able to get the maximum TSL for each 15 minutes? Or did you get the ATPC offsets that were at maximum applied during each 15 minutes? Or where the "statistics" you got more general and you thus did a more general correction. It would be good to describe that in a clearer, not necessarily longer, explanation by improving this section.
*Yes, we get the ATPC offsets that were at maximum applied during each 15 minutes. We decided to change "specifically, the modulation maxima (in dB) in the time interval" with "specifically, the maximum modulation offsets (in dB) that were applied*
120  *during each time interval" to make it stated more clearly.*

L239: "from the dataset" should be moved to the end of the sentence *Thanks.*

L244: Instead of "we let", better "we leave" *It was a typo, we meant to write "we left". Corrected, thanks.*

125

L243: "...the spatial pattern of precipitation is expected to be similar in Italian and Dutch sites (Caracciolo et al., 2006)." This is an important assumption, but I do not see how the reference proves it valid. The reference does not study spatial structure of rainfall. But your analysis indicates that the NLA, which was optimized for dutch rainfall, might lead to false-negatives when detecting small scale intermittent rainfall (see Fig 3e, or Fig 4f). This means, either the same problems must have also arisen
130  with the dutch CML data set, or the rainfall in your region has shorter spatial correlation lengths on the time scales relevant for the NLA. This should be mentioned here and discussed further when the limitations of RAINLINK are discussed.

*Thanks for spotting this referencing error. We corrected it and developed the discussion as suggested. We now give more insight in what led us to our approach and to our conclusions. We tried to keep track of the various elements which concur to the overall performance of the algorithm. We added to Section 3.2:*

"*Spherical variogram parameters (see Section 3.1, point 7. Interpolation) were calculated for three years from a pool of validated raingauges from the entire region. Range and Sill are resp. 36.12* km *and 1.12* $mm^2$. *These values resemble very much the median for May and June of the outputs of the "ClimVarParam" sub-function of Overeem et al. (2016a), which approximates 30 years of Dutch climate (van de Beek et al., 2012). Accordingly, it is expected that both the network structure and the rainfall spatial patterns are similar between the Italian and Dutch sites. This assumption drives the choice for the correct value of the NLA radius of the wet-dry classification algorithm. The differences from The Netherlands regarding orography are more relevant (see Section 2). We expect that rainfall patterns could deviate from the average behaviours described by the variograms when interacting with the complex orography of the hilly part of the region. However, we do not have enough data to calibrate the NLA radius at a small scale or considering geographical sub-samples. Moreover, a shorter NLA radius could theoretically improve the consistency with the expected decorrelation length, but, given the network in the hilly region mostly consists of medium to long links, candidates which will fall inside the NLA radius could be too few to ensure a statistical significance of the samples. Thus, we left its value unaltered at 15*km, *but we are expecting that some issues could possibly arise in the areas characterized by the most heterogeneous terrain.*"

*We then discussed if these expectations have been met in the new Discussion section (Section 5).*

Section 3.3: Some of the equations of the error metrics are not rendered correctly. Maybe they would also be more readable if they are not presented inline with the text but as separated equations. *Again the renderer failed during the merging process, we are sorry. We followed the suggestion, thank you.*

L295: "The 75% of the. . .". Do you mean 75% quantile here? Is does not seem clear from the sentence. *No, we mean literally three quarters of the total dataset. We rewrote to "75% of the 26 links' CCs are between..", hoping it will improve understandability and correctness.*

L341: It seems there are only speculations whether the "irregular and scattered precipitation patterns could be a factor that affects the correct classification". Since you are already studying this event in detail, it would be important to understand what went wrong here. Can you, with or without showing the data from the surrounding CMLs that are relevant for the NLA, elaborate on this? Would an adjustment of the NLA parameters have helped here?

*The suggestion is well posed. We did not find a less speculative explanation to date, as we lack both technical CML details and dedicated instrumentation. The speculation, however, comes from various considerations that are now described in the new Discussion section.*

L376: In my opinion a change of ME from -0.26 to -0.41 is significant and not "slightly worse" as stated here.
*Thanks, we removed "slightly".*

L377: It should be explained where the changes in ME stem from.
*Since we are considering a modification of the filtering criteria, we expect the difference to come from CML estimates smaller than 0.1* mm *which are associated with ERG5 values bigger than 0.1* mm. *Data belonging to this interval are exactly the ones which contributes to the low POD value over the same 0.1* mm *threshold, so such change seems totally reasonable.*

L379: In my opinion the ME is not in "good agreement". Could a better POD in the dutch cases be the cause for the large difference in ME?
*We reworded some statements to be less optimistic in the evaluations. Surely a better POD will mitigate the underestimation present in our dataset, but we would not call it a "cause", but another complementary indicator of the different performances of the network.*

[Figure]

**Figure 1.** Map of the Emilia-Romagna region in Northern Italy (grey area). The coloured areas are the two Provinces where the CML estimates are computed (the colour scale represents the Link Coverage, $LC$, where orange is not a negative value but exactly zero) and black thick lines delimit the two river basins (Parma, to the east, and Reno). Blue dots and red crosses indicate operational raingauge and weather radar locations, respectively, while red circles are the 100 km radar coverage. Thin black lines show two elevation contours (300 and 800 m a.s.l.). The capital cities of the two areas (Bologna and Parma, resp. BO and PR) are indicated with the black diamonds.

180    L486: "regardless the type of precipitation". While the k-R relation is for sure less sensitive to DSD variations than the Z-R relation, its robustness is only true for liquid precipitation. For larger hydrometeors like in graupel, hail or snow, it will also show a lot more scatter than for rain. The Z-R relation might be worse, but CMLs can suffer from additional large WAA e.g. because of ice or wet snow on the antenna covers. Hence, I would not say that CMLs are robust "regardless the type of precipitation".

185    *Thanks for spotting the inconsistency. What we meant was "rainfall types and regimes", not hydrometeors of different phases, so the sentence was wrong indeed. We reworded accordingly: "regardless the different types of rain (convective, stratiform, mixed)". The sentence has been moved in the Discussion section.*

Figure 1. It is good to now be able to see the grids without CML coverage in the target regions, even though, as you explain, 190    their impact is small. But now we have two types of white grid cells in the plot. A simple solution would be to use two different grey tones (light for the region, darker for LC=0) or to just adjust the limits of the virids colorbar.

*We changed the colour to the LC=0 grids from white to orange to avoid this ambiguity, both in the map and in the colour bar, where LC=0 is indicated by the negative values.*

195    Figure 3. As already stated in my first revision, the meaning of the pink horizontal band, explained in the text, should also be explained in the figure caption.

*The caption of Fig. 3 now reads: "Single link analysis for Sant'Agata (from 11.05-19:00UTC to 12.05-03:00UTC) and Vergato (from 09.06-06:00UTC to 10.06-12:00UTC): (a) and (d) show the received signals ($P_{max}$, blue; $P_{max}^{Cor}$, light blue; $P_{min}$, green; $P_{min}^{Cor}$, light green; $P_{ref}$, cyan); (b) and (e) show maximum attenuations (red), minimum attenuation (orange), estimated* 200    *rainrate (purple), and gauge measurements (black); in (c) and (f) the cumulated raingauge rainrate (black) is plotted with the*

*link estimates. Grey vertical bands correspond to intervals labelled as dry by the NLA classification,* **pink horizontal bands correspond to the threshold in** $\mathrm{dB\ km^{-1}}$ **of the Wet Antenna correction of 2.3** dB. *Y-axes ranges are specific for each CML as received powers differ between different path lengths."*

Commercial Microwave Links as a tool for operational rainfall monitoring in Northern Italy

Roversi G., Alberoni P. P., Fornasiero A. and Porcù F., July 2020

**Revised manuscript with track changes**

[revised manuscript text omitted]